# Two-thirds of global cropland area impacted by climate oscillations

Matias Heino [1], Michael J. Puma[2,3], Philip J. Ward [4], Dieter Gerten[5,6], Vera Heck[1,5], Stefan Siebert [7,8] & Matti Kummu [1]

The El Niño Southern Oscillation (ENSO) peaked strongly during the boreal winter 2015–2016, leading to food insecurity in many parts of Africa, Asia and Latin America. Besides ENSO, the Indian Ocean Dipole (IOD) and the North Atlantic Oscillation (NAO) are known to impact crop yields worldwide. Here we assess for the first time in a unified framework the relationships between ENSO, IOD and NAO and simulated crop productivity at the sub-country scale. Our findings reveal that during 1961–2010, crop productivity is significantly influenced by at least one large-scale climate oscillation in two-thirds of global cropland area. Besides observing new possible links, especially for NAO in Africa and the Middle East, our analyses confirm several known relationships between crop productivity and these oscillations. Our results improve the understanding of climatological crop productivity drivers, which is essential for enhancing food security in many of the most vulnerable places on the planet.

[1] Water and Development Research Group, Aalto University, Tietotie 1E, 02150 Espoo, Finland. [2] Center for Climate Systems Research, Columbia University, NASA Goddard Institute for Space Studies, New York, NY 10025, USA. [3] Center for Climate and Life, Columbia University, Palisdes, NY 10964, USA. [4] Institute for Environmental Studies (IVM), Vrije Universiteit Amsterdam, De Boelelaan 1085, Amsterdam 1081 HV, The Netherlands. [5] Potsdam Institute for Climate Impact Research (PIK), Research Domain of Earth System Analysis, Telegraphenberg A62, 14473 Potsdam, Germany. [6] Geography Department, Humboldt-Universität zu Berlin, 10099 Berlin, Germany. [7] Institute of Crop Science and Resource Conservation (INRES), University of Bonn, Katzenburgweg 5, 53115 Bonn, Germany. [8] Present address: Department of Crop Sciences, University of Göttingen, Von-Siebold-Strasse 8, 37075 Göttingen, Germany. Correspondence and requests for materials should be addressed to M.H. (email: matias.heino@aalto.fi) or to M.K. (email: matti.kummu@aalto.fi)

The strong El Niño phase of the El Niño Southern Oscillation (ENSO, Supplementary Note 1) in early 2016[1] led to substantial crop losses in many parts of Africa, Asia and Latin America[2]. Besides ENSO, two other climate oscillations—the Indian Ocean Dipole[3] (IOD) and the North Atlantic Oscillation[4] (NAO)—are known to strongly affect hydroclimatological processes that influence crop yields worldwide[5,6]. While climate oscillations represent a substantial driver of global crop production, recent scientific advances have improved the skill in predicting their occurrence with lead-times ranging from several months up to a year[7–9]. To improve the value of these forecasts, detailed information about the impacts of climate oscillations on crop yields would be an invaluable contribution towards ensuring food security, especially given the close relationship between supply and demand in the global food system[10].

Many connections between these climate oscillations and crop yields have been previously identified. A global study shows that ENSO has a significant influence on maize, soybean, rice and wheat yields in large parts of South Asia, Latin America and Southern Africa[11]. Local case studies have identified ENSO's impacts on crop production in China[12], the United States[13], Zimbabwe[14], Argentina[15] and Indonesia[16]. Furthermore, links between major historical ENSO events and agricultural disruptions have been documented[17], all of which had indelible impacts on past societies. Although they have not been studied as extensively, IOD and NAO are also known to influence crop yields in many areas. NAO has been found to affect crop yields in North and East Europe[6], as well as in northeast China[18] and eastern parts of the United States[6], while IOD has shown especially strong influence on Australian wheat yields[5].

Although knowledge of regional connections between these oscillations and food crop production already exists, global studies are few, and they have focussed solely on ENSO. Here we reveal the impacts of ENSO, NAO and IOD on the productivity of 12 major crop types globally for the past five decades using a consistent framework. This greatly extends past studies among which scale, data, methodology and time span have varied. Further, unlike the global studies conducted before, we use crop productivity data—simulated by the LPJmL biosphere and agrosphere model[19]—for our analyses. This approach allows us to carry out the assessment at a sub-country scale for a vast collection of crop types. This is especially beneficial for large countries such as China and the United States, where the effects might have substantial spatial variability within each country's territory[13,18]. Furthermore, the model simulations enable us to isolate the impact of climate on crop productivity, which would

be very difficult if only observed data were used, because in reality many other factors also affect crop yields[20], such as land management[21], plant diseases[22] and conflicts[23].

The results reveal that 67% of global cropland is located in areas where one or more climate oscillations show statistically significant changes in crop productivity during their strong phases. Approximately two-thirds of global food crop production comes from these areas, which are inhabited by 68% of the global population. Further, we show that in several areas across the globe, it is possible to increase the understanding of crop productivity fluctuations by using indices of several oscillations simultaneously. Our study reveals to what extent and magnitude the different climate oscillations affect crop productivity. Thereby, it provides information that can be used for increasing resilience towards natural hazards related to these oscillations in many regions across the globe.

## Results

**Crop productivity during strong oscillation phases.** The interannual variability of simulated crop productivity shows considerable spatial variations (Fig. 1b). The mean coefficient of variation (CV) of annual crop productivity globally is 0.12. Larger-than-average variation is found for Australia, southern South America and some parts of Africa as well as Russia, Canada and Central Asia. To assess how well interannual variability in crop productivity is explained by the chosen climate oscillations, we first compared the simulated crop productivity during years with strong deviations in these climate oscillations with all years. Second, we identified the sensitivity of crop productivity to variations in the oscillation indices, and finally we examined whether and where crop productivity is influenced by multiple oscillations simultaneously (see Methods section).

Of the oscillations studied, ENSO and NAO show the strongest relationships with crop productivity globally. Over 28% (38%) of global cropland is located in Food Producing Units (FPUs; see Methods section) with significant anomalies related to ENSO (NAO), populated by 1.5 (2.2) billion people (Table 1). IOD also has considerable impact in many parts of the world, as it affects crop productivity in >10% of global cropland area, inhabited by >500 million people. The negative (positive) phases of the oscillations were determined by assessing whether the yearly value of the oscillation index is smaller (larger) than the 25th (75th) percentile of the yearly index values over the whole study period. The statistical significance of the crop productivity changes during these phases was evaluated by bootstrapping ($n = 10,000$) at a 90% significance level (see Methods section).

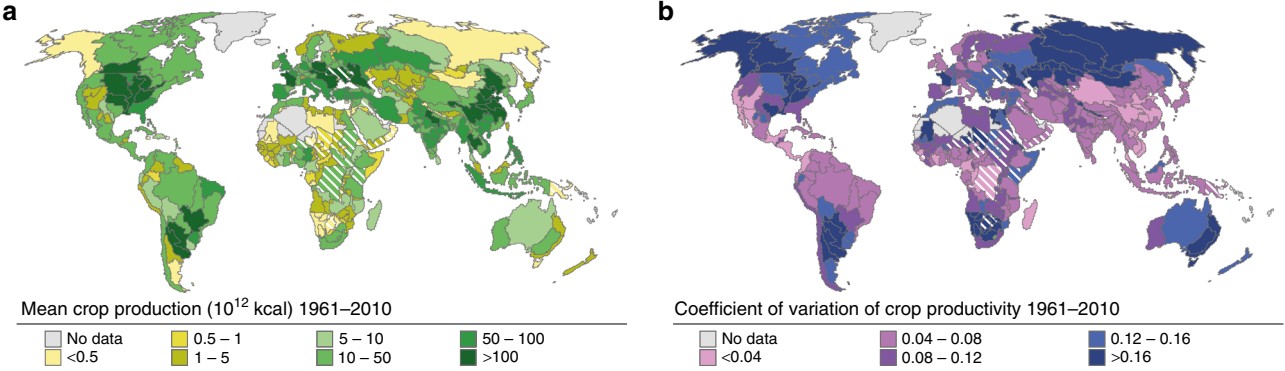

**Fig. 1** Crop production variability. **a** Total crop production and **b** coefficient of variation (CV) of crop productivity during years 1961–2010. Note that the calculation of crop production is based on the physical cropland extent[57] and crop shares[58] for each grid cell, i.e., not the harvested areas and multiple-cropping, thus leading to smaller total production compared to FAOSTAT data. The areas where Pearson's correlation between reported and simulated crop productivity is not significant ($p > 0.1$) are striped

At the globally aggregated level, only the positive phase of IOD shows a significant ($p < 0.1$) relationship with crop productivity (+1.5%). However, on a regional level (world divided into 12 regions, see division in Supplementary Fig. 1), significant anomalies were detected in two or more regions for all the studied oscillations (Fig. 2). The regions in which crop productivity is most influenced by the oscillations are North and Southern Africa as well as the Middle East, all showing significant anomalies during three different oscillation phases. Regions with weakest relationships are Central America, Eastern Europe and Central Asia, where none of the studied oscillation indices are related to significant changes in crop productivity. However, for some regions (e.g., Eastern Europe and Central Asia) anomalies of different signs can be found during the same

oscillation phases when assessed at the sub-national scale (Fig. 3), which likely cancel each other out at the country and regional scale. Furthermore, Fig. 2 shows major positive (negative) changes produced by the negative (positive) phase of IOD in Australia and Oceania.

At the FPU level, opposite phases of the oscillations often produce opposite changes in crop productivity (Fig. 3). The strongest changes related to ENSO occur in the northern and southern parts of Africa, India, Southeast Asia and northern portions of South America (Fig. 3a, b). IOD shows an especially strong influence on crop productivity in eastern Australia, but anomalies related to IOD can also be found in some parts of central and Southern Africa (Fig. 3c, d). In addition to ENSO and IOD, NAO also shows a strong effect on crop productivity in many parts of Africa, which is consistent with the findings shown in Fig. 2. Furthermore, NAO affects crop productivity in Europe as well as in parts of eastern China and South America (Fig. 3e, f). In FPUs where the correlation between reported and simulated crop productivity is insignificant (see Methods section), the results should be treated with caution (these areas are striped in Figs 1, 3–5).

**Sensitivity of crop productivity to oscillation indices.** We also assessed the sensitivity of crop productivity to interannual variations in the oscillation indices. This is important, because the above examination of the anomalies only reveals the response of crop productivity to strong phases in the climate oscillations (Fig. 3), while in sensitivity analysis we examine the relationship of crop productivity also with smaller changes in the oscillation

| Table 1 Globally aggregated impacts of ENSO, IOD and NAO | | | |
|---|---|---|---|
| Oscillation | Population ($10^9$) | Cropland ($10^6$ km$^2$) | Crop production ($10^{15}$ kcal year$^{-1}$) |
| ENSO | 1.5 (28%) | 4.1 (28%) | 2.2 (28%) |
| IOD | 0.6 (11%) | 2.0 (14%) | 0.7 (9%) |
| NAO | 2.2 (41%) | 5.6 (38%) | 3.3 (41%) |
| Any | 3.7 (68%) | 9.8 (67%) | 5.2 (65%) |

Population, cropland, and average total crop production (and percentage) in the areas where one or more oscillations produce significant changes in crop productivity during their negative and positive phases, when assessed at the sub-national (i.e., FPU) scale. Only the areas where Pearson's correlation between reported and simulated crop productivity is significant ($p < 0.1$) are included. Unmasked results are provided in Supplementary Table 4

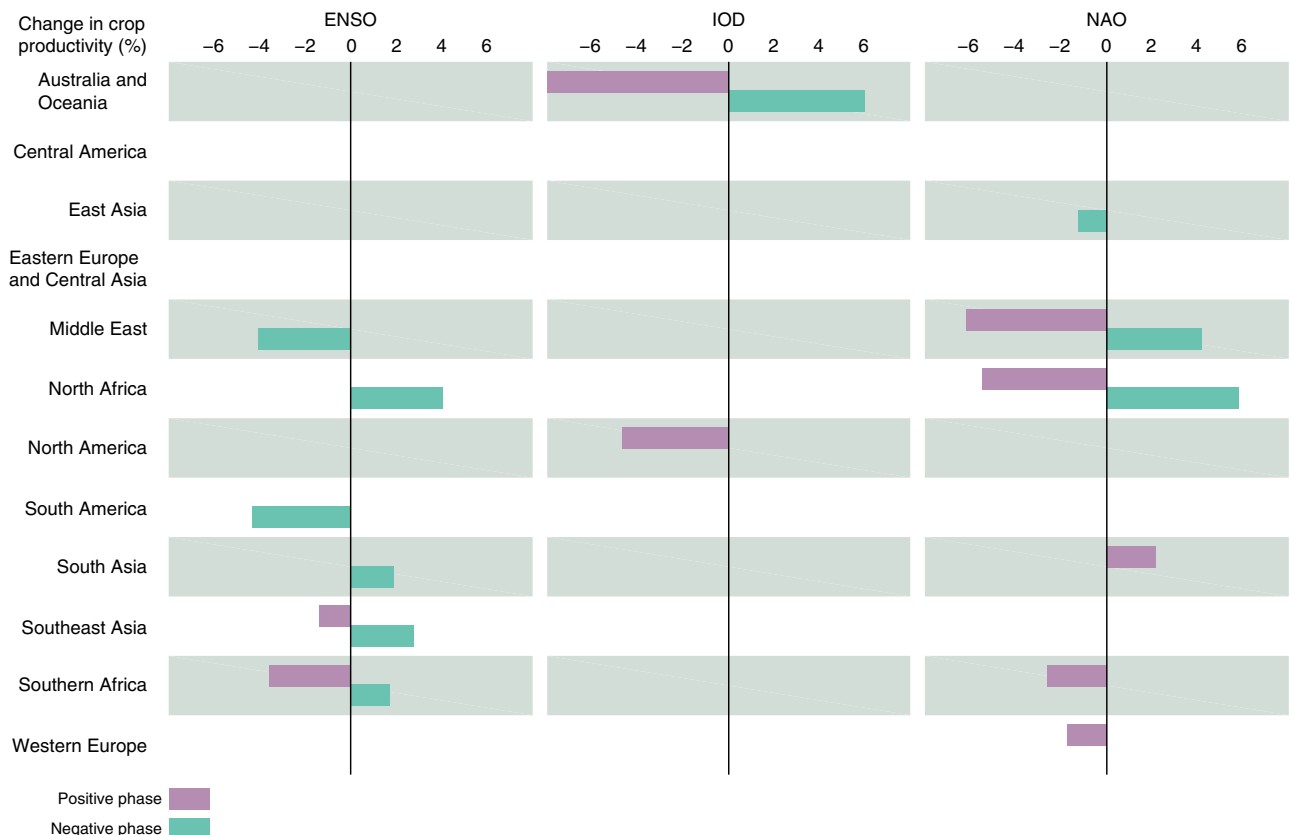

**Fig. 2** Crop productivity changes during strong oscillation phases at the regional scale. The years with negative (positive) oscillation phases were determined by inspecting whether the yearly value of the oscillation index is smaller (larger) than the 25th (75th) percentile of the yearly index values over the whole study period. The analysis was conducted by first aggregating crop productivity data to the regional scale and then assessing the changes (see Methods section). Only significant changes are indicated, as computed by bootstrapping ($n = 10,000$, $\alpha = 0.1$). See tabulated results in Supplementary Table 1

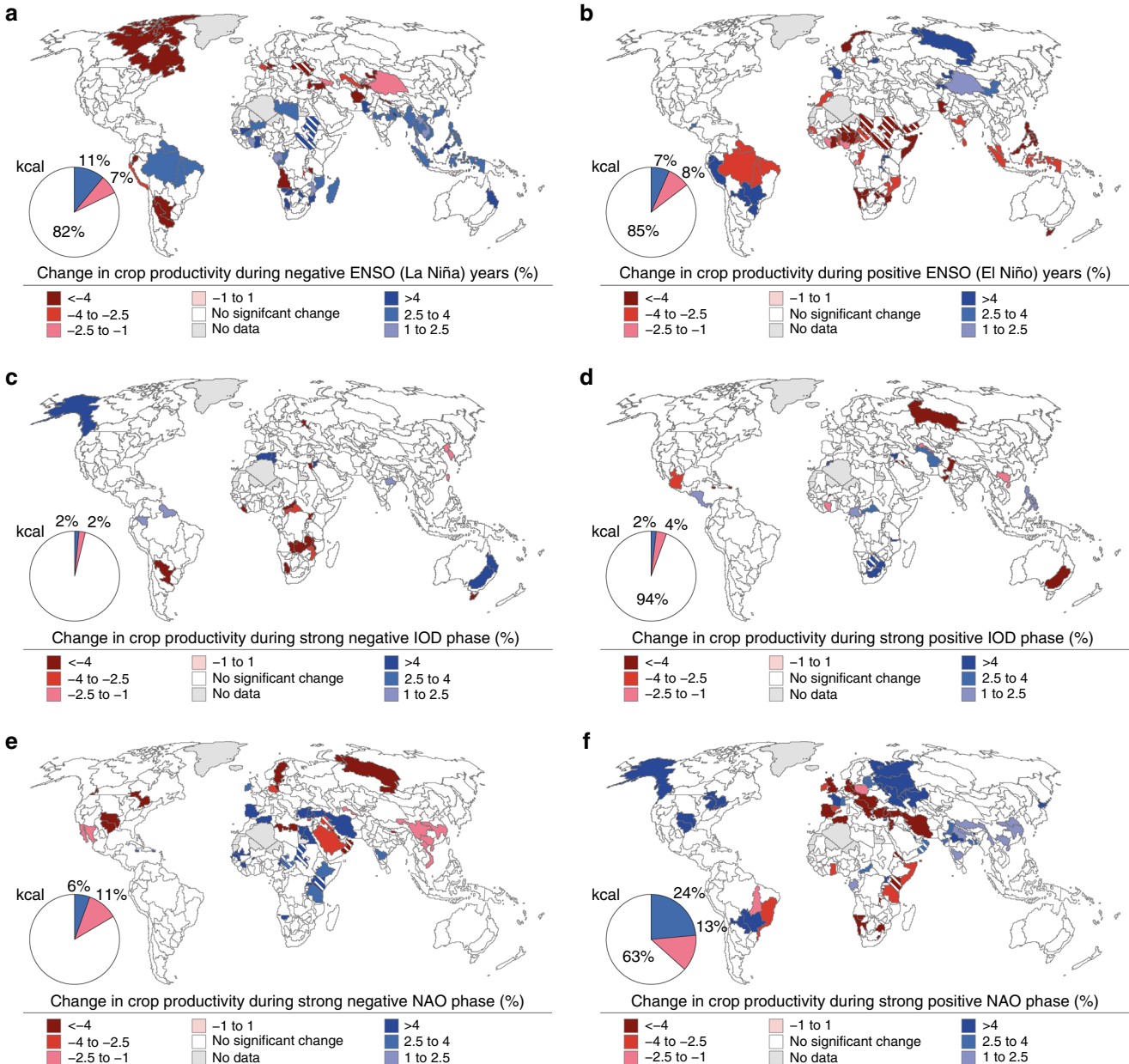

**Fig. 3** Crop productivity changes during strong oscillation phases at the FPU scale. Changes in crop productivity, at sub-national (i.e., FPU) scale, during strong oscillation phases of **a**, **b**, ENSO; **c**, **d**, IOD; and **e**, **f**, NAO. The significance of the changes was assessed with bootstrapping ($n = 10,000$, $\alpha = 0.1$). The pie charts show global proportions of crop production in areas with significantly increased (blue) and decreased (red) crop productivity during the different phases of the oscillations, while the pie slice in white is the proportion of global crop production that does not show a significant change. The areas where Pearson's correlation between reported and simulated crop productivity is not significant ($p > 0.1$) are striped and not included in calculating the global proportions shown in the pie charts

indices (Fig. 4). This analysis was conducted by linear regression, in which the slope represents the rate of change in crop productivity per unit change in the oscillation indices.

We find that at the regional level crop productivity is influenced by at least one of the oscillations (ENSO, NAO or IOD) in 8 out of the 12 regions (see Supplementary Table 2). Crop productivity is insensitive (no significant correlation, see Methods section) to the studied climate oscillations only in Eastern Europe and Central Asia, North America, Western Europe and Central America. Especially sensitive regions to the climate oscillations in terms of crop productivity are North (ENSO and NAO) and Southern (ENSO) Africa, Southeast Asia (ENSO) as well as Australia and Oceania (IOD).

On a sub-national scale (Fig. 4), the crop productivity in Southeast Asia and many parts of Africa is especially sensitive to variations in the ENSO index (Fig. 4a, c). Some of the results in Africa should be treated with caution, as simulated crop productivity does not correlate ($p > 0.1$) with reported crop statistics. Furthermore, the ENSO index also correlates with crop productivity in the northern parts of North and South America (Fig. 3a). Again, the IOD index correlates best with crop productivity variations in Australia (Fig. 4b). Furthermore, IOD's influence on crop productivity is found for India and Southern Africa. NAO appears to have strong effects on crop productivity in many parts of Europe, Middle East and China as well as some parts of Africa and Northern India (Fig. 4c). The opposite

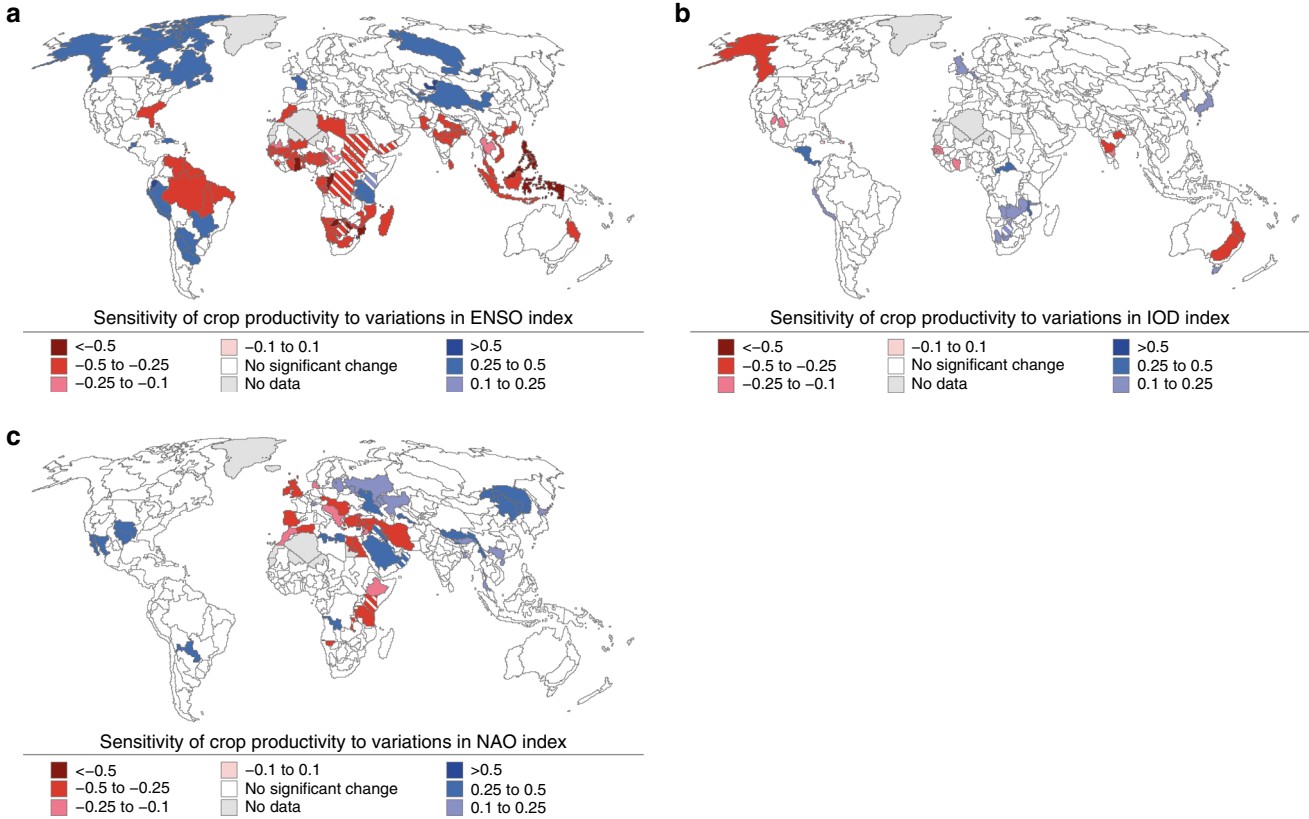

**Fig. 4** Crop productivity sensitivity to oscillation indices. Sensitivity of crop productivity to changes in **a** ENSO, **b** IOD, and **c** NAO indices at the FPU level. The linear relationship was concluded to be significant based on the $p$-value (<0.1, parameterized, see Methods section) of Pearson's correlation coefficient. Note that the areas where Pearson's correlation between reported and simulated crop productivity is not significant ($p > 0.1$) are striped

sensitivities to NAO in Europe (negative in west, positive in northeast) might explain the fact that no significant sensitivity is found at the regional scale (cf. Supplementary Table 2).

We present three plots showing the relationships between the sensitivity of crop productivity to the oscillations and the CV of crop productivity in Supplementary Fig. 16. No significant positive correlation exists between sensitivity and CV for any of the oscillations, which suggests that globally other climatological factors also influence variability in crop productivity. However, Supplementary Fig. 16 still illustrates a discernible global effect of the oscillations on crop productivity: 27% of global crop production is sensitive to variations in ENSO, 5% to variations in IOD, and 20% to variations in NAO. To obtain a preliminary understanding of the dynamics in the relationship between crop productivity and the oscillations over time, we also conducted a moving window (Spearman's) correlation analysis and assessed whether a trend exists in its strength (see Supplementary Methods 1). The correlations are found to be consistently significant for ENSO in Africa, Southeast Asia and northern South America (Supplementary Fig. 17). In Australia, IOD seem to have a strong impact through the whole study period, while NAO correlates consistently with crop productivity, e.g., in the Middle East.

**Combined influence of the oscillations on crop productivity.** It is known that the oscillations may amplify or weaken each other and produce combined climatological effects[24,25]. Therefore, we also studied the relationship between combinations of oscillations and crop productivity, using a multivariate regression model (see Methods). At the global scale, the explanatory power of the oscillations on crop productivity is low (adjusted $R^2 = 0.024$), with

the NAO index being the only variable showing any significance ($p$-value < 0.1). On a regional scale (Supplementary Table 3), the multivariate model can explain crop productivity best in North Africa (adjusted $R^2 = 0.39$; significant oscillations: ENSO and NAO), Southern Africa (0.31; ENSO), Southeast Asia (0.25; ENSO) and Australia and Oceania (0.22; IOD). However, the model does not show any significant correlation ($p > 0.1$) with crop productivity in North America or Eastern Europe and Central Asia.

The results at FPU scale reveal that the multivariate regression model has the highest explanatory power for crop productivity in many parts of Southeast Asia, Australia and northern parts of South America as well as Southern Africa ($R^2 > 0.3$, Fig. 5). Of the oscillations studied, only ENSO shows a significant relationship with crop productivity in northern South America as well as parts of western Africa and South Asia (Fig. 5b). Together with IOD, ENSO contributes to crop productivity variations in some parts of Southern and Eastern Africa as well as the coasts of Peru and Colombia. NAO is the most important driver of crop productivity in northern China. NAO together with ENSO drive crop productivity variability, for example, in Southeast Asia and northeastern Africa. In northeast India and northeast Australia, the variability in crop productivity can be explained best with a combination of all three oscillations.

## Discussion

The results provided in this study show that currently almost four billion people live in areas where at least one of the three major climate oscillations (i.e., ENSO, IOD and NAO) exerts a significant influence on crop productivity. Moreover, these regions

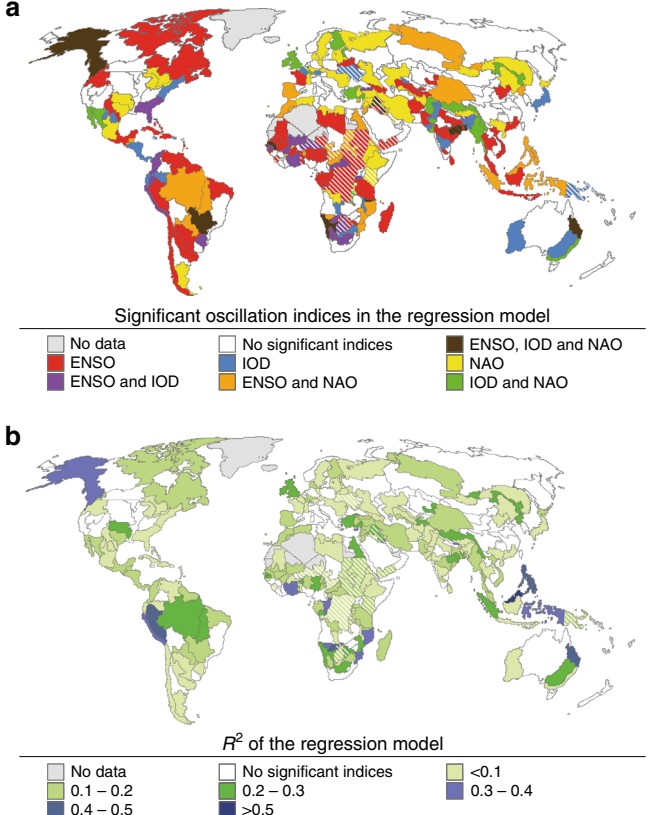

**a**

Significant oscillation indices in the regression model

| | | |
|---|---|---|
| No data | No significant indices | ENSO, IOD and NAO |
| ENSO | IOD | NAO |
| ENSO and IOD | ENSO and NAO | IOD and NAO |

**b**

$R^2$ of the regression model

| | | |
|---|---|---|
| No data | No significant indices | <0.1 |
| 0.1 – 0.2 | 0.2 – 0.3 | 0.3 – 0.4 |
| 0.4 – 0.5 | >0.5 | |

**Fig. 5** Combined effects of the oscillation indices on crop productivity. **a** shows which oscillation indices are found to be significant explanatory parameters in the multivariate regression model with crop productivity as the dependant variable, and **b** shows the adjusted $R^2$ of the model for each FPU. The areas where Pearson's correlation between reported and simulated crop productivity is not significant ($p > 0.1$) are striped

produce approximately two-thirds of the global food crop calories (Table 1). Our study extends previous knowledge by analysing the crop productivity impacts of these climate oscillations, both separately and combined, on a global scale and at the sub-national level. The analysis was conducted by using simulated crop productivity data, which allowed us to isolate the climate influence from other drivers of agricultural productivity during the period 1961–2010.

Notably, for globally averaged crop productivity, only the positive phase of IOD shows a significant change. However, its areal impact is relatively small compared to that of ENSO and NAO (cf. Fig. 3), and our other analyses do not show that variations in globally averaged crop productivity can be explained by the oscillations individually or combined (Supplementary Fig. 18). This suggests that regional crop production deficits due to these oscillations could be compensated by interregional food imports into the affected regions. For areas unable to compensate the losses through trade, these low productivity years might still be devastating[26]. This was the case, for example, in southern parts of Africa during and after the 2015–2016 El Niño event, which resulted in an estimated 12% drop in aggregate cereal production[2] and led to over 32 million people suffering from food insecurity[27]. It is important to note though that a part of the decline might have been due to increased water scarcity[28], anthropogenic depletion of water resources[29] or droughts occurring due to other factors.

We also found that the effects of climate oscillations on crop productivity are not always stationary but change over time

(Supplementary Fig. 17). According to our analyses, opposite phases of the oscillations do not necessarily produce crop productivity changes of opposite directions. This is, for example, the case in Western Europe, where a negative phase in NAO does not seem to be related to changes in crop productivity, whereas a positive phase does (cf. Fig. 2). The lack of such a relationship is likely to be the reason why the NAO index does not show a significant correlation with crop productivity at regional scale (Supplementary Table 2). Details about change and sensitivity analyses at regional and global level are presented in Supplementary Figs 18–21.

While many regional and local studies on the effects of climate oscillations on crop yields and production have been conducted, only few global studies exist. Within those studies, the methods and the time span are not fully consistent, which already in itself causes differences in the results. Furthermore, most previous studies examined only crop-specific relationships[5,11,16], whereas this study aggregated the yields (kg ha$^{-1}$) of all major crop types included in the LPJmL model to crop productivity (kcal ha$^{-1}$ year$^{-1}$).

Two global studies, which analyse the effects of ENSO on the yields of maize, rice, soybean and wheat[11] as well as an agricultural stress index[30], show similar patterns compared to this study. For example, a strong relationship between ENSO and crop productivity was identified in the northern parts of Africa, Southeast Asia, India and parts of South America in both studies, which is in line with our findings (cf. Figs 3 and 4). However, the results are not fully comparable as both studies have a different time span, and they analyse the effects of ENSO for different metrics. In India, the effects of ENSO on crop productivity (cf. Figs 3 and 4) are supported by a regional study which shows that during a La Niña (El Niño) event crop production tends to increase (decrease)[31]. Furthermore, our findings regarding the strong impacts of ENSO on crop productivity in Indonesia are underpinned by regional studies that show crop production to shift markedly due to climatological anomalies driven by ENSO[16].

In addition to ENSO, NAO has also been shown to influence crop yields in the north-eastern parts of China[18], which is in line with this study (cf. Fig. 4). Further, similar patterns to our study have been found in many parts of Europe, where NAO shows a positive correlation with vegetation production during spring[18] but negative correlation during summer[32]. Moreover, there is evidence of NAO's influence on vegetation health or yields in northern Africa, Central Asia, the Middle East[18] and the southeast United States[33]. Connections between NAO and hydrological variability have been noted for several areas in the Middle East[34] as well as parts of East Asia[35]. For IOD, a study on Australia's wheat production shows very similar patterns to our study (cf. Figs 3 and 4), as wheat production is found to decrease during the positive phase in IOD[5]. Furthermore, in Australia the positive phase of IOD has been linked to severe drought events[36], which likely affect agriculture as well. See Supplementary Note 2 for additional details regarding the climatological impacts related to the oscillations.

Our results newly indicate that NAO has a statistically detectable influence on crop productivity in areas where NAO's effects have this far been less studied, for example, in Africa and the Middle East. It would therefore be important to confirm and gain a better understanding of these results by conducting similar studies on a regional scale using more detailed local data. Furthermore, as human actions can mitigate, but also increase, the negative impacts related to climate oscillations, in future studies these results could be embedded with reported data, enabling assessment of the effectiveness of different adaptation mechanisms (e.g. fertilizer use). Using reported crop statistics could also enable analysis of the relationship between climate oscillations

and harvested areas, which is also affected by weather-related disasters[37], reinforcing the impacts on total crop production.

Furthermore, it would be of great benefit to conduct an analogous analysis for each crop separately to identify the crops that thrive under certain phases of the oscillations. Another aspect to consider in future studies is possible delays in the effects of the oscillations on crop productivity, i.e., time-lagged correlations, as identified, e.g., in a previous study by Wang & You[18].

Results of this study rely on one model only, yet earlier studies found that the magnitudes and spatio-temporal variation of LPJmL-simulated crop yields are comparable to simulations from other global crop models and to reported statistics[38–40]. As a part of the Agricultural Model Intercomparison and Improvement Project, Müller et al.[38] evaluated yields of wheat, maize, rice and soybean simulated by 14 global gridded crop models (GGCM). Müller et al.[38] showed that LPJmL performs as well as other GGCMs in reproducing reported yields, being the model that performs best for wheat. While none of the GGCMs studied by Müller et al.[38] perform well in rice yield simulations (in LPJmL likely because of limited account of multiple cropping[19]), Frieler et al.[39] showed that LPJmL follows the GGCM ensemble mean for rice yield. Another study demonstrates that LPJmL can clearly separate effects of heat and drought stress on temporal variation in crop yields (soybean, maize, wheat)[40].

While all GGCMs are subject to uncertainty related to factors independent of model skill, including harvested areas[41], soil data[42] and climatological input[43], it is important to note that crop statistics used for evaluation are also inherently uncertain in some regions. For example, the two reference data sets of reported crop statistics[44,45] exploited by Müller et al.[38] did not correlate well everywhere. In addition, such reports usually refer to the production on the area harvested[46], ignoring thereby effects of complete crop failure and potentially underestimating the variability in crop yields caused by climate extremes. The crop statistics are especially unreliable in many African countries[47], which is also where the comparisons of this study show lowest match between observed and simulated crop productivity (Supplementary Figs 6–15, see Methods section). In general, the agreement between different reported data sets is better in high input agricultural regions, where reported crop yields have noted to be susceptible to climate variability[20,37].

We emphasize that in this study we only assessed the effect of climatic variability on crop productivity. However, many non-simulated factors (e.g., management decisions, pests and conflicts) also affect reported crop statistics, and their impact can be even more severe than that of climate variability[20,22,23], depending on year and region in question. Given that only a third of global crop yield variability can be attributed to climate variability[20] and that our analysis is focussed on the isolated climate effect, it is not to be expected that it reproduces all observed crop yield dynamics.

In view of anticipated climate change and population growth, increasing the resilience of agricultural production is imperative[48]. Our spatially explicit findings reveal the extent and magnitude to which variations in crop productivity are influenced by climate oscillations. This information might inform policymakers for increasing resilience towards natural hazards—especially in regions like Australia, China, Southeast Asia and some parts of Africa, where this analysis verifies former evidence that these regions are particularly prone to the impacts of these oscillations. Furthermore, our results suggest that it is principally possible to increase the contribution and value of forecasting climate oscillations towards ensuring food security by using information from multiple oscillations simultaneously. The FAO provided a global action plan to tackle agricultural vulnerability to the 2015–2016 El Niño event[49], and, for example, in Somalia, the preparedness

towards the El Niño of 2016 prevented crop losses worth millions of dollars by actions (e.g., polypropylene bag and tarpaulin sheet distribution) allowing farmers to prepare for expected flooding[50]. Our results are therefore an important step towards understanding the role of climate oscillations on crop productivity around the world, which is potentially useful for improving local disaster control in the most vulnerable places on the planet.

## Methods

**Model set-up.** The crop productivity data for this research were derived from the LPJmL model[19,51,52], which has a proven record of quantifying, e.g., effects of climate variability on crop yields[38–40]. LPJmL is a process-based global dynamic vegetation and hydrology model able to capture the production of the most common natural (9) and agricultural (12) vegetation types in coupling with associated biogeochemical processes, carbon and water fluxes. In the model, temperature controls crop growth through sowing dates, phenological development and potential evapotranspiration, which together with precipitation and/or irrigation controls soil moisture (i.e., water availability for crop growth). The crop-specific annual yields (kg ha$^{-1}$ year$^{-1}$) of the here considered 12 major crop types were converted to caloric crop productivities (kcal ha$^{-1}$ year$^{-1}$), after which the total crop productivity was calculated as the growing area weighted mean.

The LPJmL model was run at a resolution of 0.5°; results were then aggregated to the scale of FPUs. FPUs divide the globe into 309 areas based on trade and water management[53,54]. This enables analysis on a sub-national scale specifically designed for food production and climate-related studies. This is especially beneficial for analysis of large countries such as the USA and China, where the effects of the climate oscillations may vary between different parts of a country[18,55].

To isolate the effect of climate variability on crop productivity, only climate inputs were allowed to vary in the simulations. All other parameters and inputs—such as land use, agronomic practices and $CO_2$ concentration—were kept constant at year ~2000 values[56]. An important note regarding the total crop production data (e.g., Table 1, Fig. 3) is that the cultivated areas[57] and crop shares[58] for each 0.5° raster cell are physical—not harvested—areas, which prevents accounting for multiple-cropping, thus leading to smaller total production compared to FAOSTAT data. The climatological daily forcing data were obtained from the Global Meteorological Forcing Dataset for land surface modelling provided by Princeton University, spanning 1948–2010[59]. The use of daily forcing data ensured that the model is able to capture the influence of timing and duration of, e.g., heat and drought on crop productivity.

**Calibration and evaluation of the model.** The calibration of agronomic practices in the LPJmL model was conducted by adjusting, for each crop type and country, the maximum leaf area index (values between 1 and 7), the harvest index (the maximal fraction of above-ground biomass allocated to storage organs at harvest in the absence of water stress) and the radiation use efficiency (scaling factor for the conversion of intercepted photosynthetically active radiation into biomass) so that the simulated yields best match reported country-level crop yield statistics[52]. Not all management interventions are currently being modelled (including fertilizer application), making these adjustments necessary in order to ensure adequate simulation of crop yields. For countries where no country-level yield data existed, but the land-use data had areas allocated to the crop type, the yields were calculated assuming a moderate maximum leaf area index of 5. Comparisons between reported and simulated yields for years 2001–2010 and 1981–1990 are shown in Supplementary Figs 2–5. Generally, the calibrated crop yields represent reported crop statistics well: The Willmott coefficient calculated between reported and simulated yields is above 0.9 for most crops and varies between 0.66 (sugarcane) and 0.99 (wheat).

To compare the simulated crop productivities to observations, two data sets of reported crop statistics were employed: annual data of reported crop yields and harvested areas of maize, rice, soybean and wheat at a 0.5° spatial resolution[20], and reported annual data of all crops included in LPJmL at the country scale from FAOSTAT[46]. Further, to compare LPJmL simulation results with reported crop statistics and to roughly assess whether adding other than climatological signals to the simulated data affect the results, a second simulation was conducted. In this simulation, agronomic practices were calibrated decennially and land use as well as $CO_2$ concentration were also allowed to vary historically, similarly to Porkka et al.[60].

In a first step of this evaluation, Pearson's correlation was calculated between simulated (decennially calibrated) and reported crop productivity. Second, Pearson's correlation was calculated between simulated (calibrated for year 2000) and reported (de-trended by fitting and subtracting a best-fit polynomial curve) crop productivity. Third, a comparison of the standard deviation and the CV between simulated (calibrated for year 2000) and reported (de-trended) crop productivity was conducted by calculating the non-parametric Spearman's correlation (alternatively to Pearson's correlation to account for the large variation in the tested variables) between these data. Additionally, the country-level

comparisons in steps one and two were also conducted separately for each of the 12 crop types simulated by LPJmL.

The correlations are found to be high (Pearson's $r > 0.5$) in most parts of the world for decennially calibrated simulated crop productivity (Supplementary Figs 7 and 8). However, in some areas correlations are insignificant ($p$-value > 0.1)— usually in regions with low crop production, such as central Africa. Further, reported (de-trended) crop productivity also correlates ($p$-value < 0.1) positively with simulated (calibrated for year ~ 2000) crop productivity in many countries (Supplementary Fig. 9a) and FPUs (Supplementary Fig. 9b) around the world. This is noticeable, especially considering that non-climatic factors extensively influence reported crop productivity variability[20,22,23]. In the crop-type-specific analyses, LPJmL-simulated wheat productivity follows the patterns of reported wheat yields in most parts of the world, while sugarcane shows less agreement with reported data (Supplementary Figs 10 and 11). Also, the standard deviation and CV of crop productivity show similar patterns between simulated (calibrated for year 2000) and reported (de-trended) crop productivity (correlation $p$-value for both <$10^{-5}$; Supplementary Figs 12 and 13). On average, the CV is slightly higher for the reported crop productivity; this is the case, for example, in parts of Southern Africa (Supplementary Figs 14 and 15), potentially because of strong influence of non-climatological factors on crop productivity fluctuations.

Furthermore, to compare the sensitivity of the results to the chosen model set-up, two additional model runs and analyses were performed. Prior to a sensitivity analysis on the decennially calibrated simulated crop productivity time series (see details above), it was de-trended by fitting and subtracting a best-fit polynomial curve from the original data. This analysis was conducted in order to evaluate whether adding other than the climatological signals to the simulated crop productivity data would affect the results. A second sensitivity analysis was conducted based on crop productivity simulated with limited irrigation water supply (i.e., restricting water use by renewable water availability). This simulation was performed in order to understand the possible effect of the irrigation set-up in the LPJmL model on the results of this study, as in the main simulations crop productivity is not limited by water availability in irrigated areas. Supplementary Figs 22 and 23 show the results of the sensitivity analysis of these simulations; the patterns remain similar for both compared to the original sensitivity analysis (cf. Fig. 4).

**Oscillation indices**. To represent the variability of ENSO, IOD and NAO, three indices were chosen: the Japan Meteorological Agency SST Index[1], the SST Dipole Mode Index[3,61] and Hurrell's North Atlantic Oscillation Index (PC-based)[4,62], respectively. All calculations were conducted for each index separately except for the multivariate regression, which included all indices simultaneously. We chose these specific indices as they are all well established and have already been used in numerous studies related to crop production as well as hydrology[5,6,63]. They were transformed to yearly values by calculating the index means for months when the oscillations tend to have the strongest signal (i.e., NDJ for ENSO, NDJF for NAO and SON for IOD)[3,64,65]. A more detailed description of the indices can be found in Supplementary Table 5.

In order to analyse whether the results are robust to the chosen oscillation indices, the sensitivity analysis (see below) was also conducted for four other indices: the Oceanic Niño Index (ENSO)[66], the Southern Oscillation Index (ENSO)[67], the SLP Dipole Mode Index (IOD)[68], and the Hurrell North Atlantic Oscillation Index (station-based)[69]. Although some differences occur (cf. Fig. 4, Supplementary Fig. 24), these analyses largely confirmed the patterns found for the indices selected in this study.

**Crop productivity during strong oscillation phases**. The crop productivity anomalies occurring during strongly oscillating years were defined as a change (in percentage) between average crop productivity during those years and average crop productivity of all years. Strongly negative (positive) phases of the oscillations were assumed to prevail in years (listed in Supplementary Table 6) when the respective oscillation index was smaller (larger) than the 25th (75th) percentile of all yearly index values. The statistical significance of the changes was assessed by boot-strapping ($n = 10,000$) at a 90% significance level. The bootstrapping procedure was conducted so that the mean change was calculated for each bootstrap sample. If over 95% (two-sided test) of the sample of means were either larger or smaller than zero, the change was considered statistically significant. To analyse how sensitive the results are to the chosen significance level, the analysis was also conducted at 5 and 20% significance levels. With a 5% (20%) significance level, the total extent of cropland areas that would be affected by the studied climate oscillations decreased (increased) by 15% (18%).

**Crop productivity sensitivity to oscillation indices**. The sensitivity of crop productivity to variations in the oscillation indices was assessed by correlation analysis and linear regression. Prior to these analyses, all data were standardized (except the IOD and NAO indices, which are already standardized by definition), to make the results comparable between different regions and indices. Then, a least squares linear regression was fitted to the standardized data. The slope of the linear fit represents the sensitivity of crop productivity to variations in the indices. The statistical significance of a linear relationship was assessed by calculating the $p$-value ($p < 0.1$) of Pearson's correlation between the oscillation index and crop

productivity. The $p$-value was calculated by transforming the correlations using Student's $t$-distribution.

**Combined influence of the oscillations on crop productivity**. We also examined whether there are regions where the variability in crop productivity can be best explained by several indices simultaneously with a multivariate linear regression model. The model includes all indices as explanatory variables and the regression was conducted by the least squares method. The results of this analysis show: (a) which oscillation indices are found to be significant explanatory parameters in the multivariate regression model; and (b) the adjusted $R^2$ of the model for each FPU. The significance ($p < 0.1$) of each index as explanatory variables was assessed by calculating the $t$-statistic for each index.

Before the actual regression calculations, a few modifications were made to the data. First, the oscillation indices were scaled to have a range between 0.01 and 1.01. Second, the crop productivity data were standardized. Finally, a link function was applied to the standardized oscillation indices. A list of the link functions that were applied for each index and FPU can be found in Supplementary Table 9. These modifications were conducted in order to address the weaknesses of the least squares method in fitting non-linear data. To ensure that the regression works correctly, the oscillations were inspected for multicollinearity, which was not detected. The inspection was conducted so that Pearson's correlation was calculated between each index pair as well as calculating the variance inflation factor (<4; Supplementary Table 7 and Supplementary Table 8).

**Masking the results based on model performance**. To account for uncertainty in the simulated crop productivity data, we masked the results in areas where <15 years of reported country level data were available or where the correlation between simulated (decennially calibrated) and reported country-scale crop productivity is insignificant ($p > 0.1$) or negative. The masking was transformed from country scale to FPU scale, so that each FPU was attributed the masking status of the country with the largest areal within an FPU in question. For most FPUs, the transformation was very straightforward, as they are constructed as a merge between river basins and administrative regions. In map figures, the results of the masked FPUs are striped (e.g., cf. Fig. 1) and all of the aggregated (i.e., global and regional) results were calculated so that the masked FPUs were omitted from the calculations. However, the masking does not have much influence on the aggregated results, e.g., see the global results with (cf. Table 1) and without (Supplementary Table 4) masking.

**Changes in crop productivity response to the oscillations**. The variability in crop productivity responses to the oscillation indices was assessed by calculating the Spearman's correlation coefficient with a 21-year moving window. Owing to the variability of the available oscillation index data, the length and the timing of the analyses vary. The first correlation window for ENSO, NAO and IOD data is 1961–1971 and the last window is 1990–2010 for ENSO and NAO and 1989–2009 for the IOD. Prior to the trend assessment, the proportion of years when the moving correlation windows have a significant Spearman's correlation ($p < 0.1$) was calculated. In the trend assessment, the changes in the relationship were divided into six categories, which are: no significant correlation in any years, no trend, weakening trend, strengthening trend, correlation shifts from negative to positive, and correlation shifts from positive to negative. Trend existence and whether a trend is positive or negative was assessed using absolute correlations values. Further, the criteria for correlation to have shifted from negative (positive) to positive (negative) was that a positive (negative) trend exist and that both significant negative and positive correlations are found in the correlation time series. The Mann–Kendall trend test ($p < 0.05$) was used to assess whether a trend exists. See the motivation for this analysis in Supplementary Note 3 and the results in Supplementary Fig. 17.

**Data availability**. The crop production data from the LPJmL simulations and the harvested areas data used for data aggregation (at FPU and country scales) have been deposited in the Dryad Digital Repository[70] (https://doi.org/10.5061/dryad.6h5p0).

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

## Acknowledgements

This study was funded by *Maa- ja vesitekniikan tuki ry*, Emil Aaltonen foundation ('eat-less-water' project), Academy of Finland SRC project Winland, and Academy of Finland projects SCART (grant no. 267463) and WASCO (grant no. 305471). Additionally, M.J.P. gratefully acknowledges support from the Columbia University Center for Climate and Life, where he is a Climate and Life Fellow, and from the Interdisciplinary Global Change Research under NASA cooperative agreement NNX14AB99A supported by the NASA Climate and Earth Observing Program. P.J.W. received funding from the Netherlands Organisation for Scientific Research (NWO) in the form of a VIDI grant (grant no. 016–161–324). S.S. acknowledges funding from the Federal Ministry of Education and Research (BMBF) for the project GlobeDrought (grant no. 02WGR1457A) through the Programme Global Resource Water (GROW). We are grateful for Dr Timo Räsänen, Dr Joseph Guillaume, Professor Olli Varis, Dr Deepak Ray and Sibyll Schaphoff for their support and help in conducting the study.

## Author contributions

M.H. and M.K. designed this study in consultation with M.J.P., P.J.W., D.G. and S.S. LPJmL model runs were performed by V.H. Analyses and statistics were conducted by M.H. supported by all co-authors. M.H. wrote the article, with contributions from all co-authors.

## Additional information

**Competing interests:** The authors declare no competing financial interests.

