## [Peer Review File(PDF 971 kb) · Nature Communications]

Reviewers' comments:

Reviewer #1 (Remarks to the Author):

This is basically a modeling study in which the authors related modeled crop production to observed climate oscillation indices. Ideally such studies should be conducted using the observed / reported crop production information. Irrespective of how sophisticated the methodology is, the fact remains that the result hinges on unverifiable crop production information.

How do the authors know that the simulated food production using LPJmL model is accurate? They have not been used to compare against historical crop yields, and the significantly more tricky production information (because to get the production correct one needs to accurately know the harvested area information as well). No amount of modeling can ever get the harvested extent correct, as that is discretion of farmers and thus only reported information can give an accurate picture. So, I am having difficulty in accepting the results other than that this is a modeling study so may or may not be accurate in each of the food production units.

Now the authors do indeed try to make the argument that the correct approach is indeed to use modeled crop yields (not production really because the authors can never simulate the harvested areas) but land up contradicting themselves. If I understood the argument correctly, climate oscillations impact food production; but if observed data were used then the signal from climate oscillations would not be captured. And therein lies the contradiction - if the signal of climate oscillations is not present in the observed crop data that basically means that there is no impact - there is either really no impact or there is a coping mechanism.

As I have noted above - production information requires two independent pieces of information - crop yields and crop harvested areas. The later is quite impossible to simulate. Even the former - crop yields from the LPJmL model has not really been validated to such a level of accuracy as to allow its use in isolating such a nuanced signal of climate oscillation on crop yields. There are several publications that was referred to show the modeled yields. I went through each of them to understand how well the modeled yields compared with reported yields as simulated by the author's model of choice LPJmL.

Fader et al., 2010 compared LPJmL simulated yields to FAO yields for year 1999 to 2003 with substantial errors in major crop countries even when simulating maize globally, and barley, rye, wheat, crops but only in temperate regions. So, I conclude that this paper's comparison / validation does not give me confidence to use for the period 1950-2006.

Bondeau et al., 2007 simulated temperate cereals and maize and determined average yields over 1991 to 2000 and compared the simulated yields with FAO reports. So this report also cannot be used for conducting a study of food production from 1950 to 2006.

Kummu et al., 2014 kept the food part of their study fixed at year 2000.

Schaphoff et al., 2013 does not concern with crop yield / production but rather carbon using the LPJmL model.

Gerten et al., 2011 held their crop production information fixed at year 2000.

Rockström et al., 2009 was not concerned with simulating crop yields / production but on water availability.

Rosenzweig et al., 2014 presented the results of intercomparing global crop model simulations out of which LPJmL is one model and all the models had disagreements in simulated crop yields to various amounts.

Fader et al., 2013 is a study of future scenarios but does have a baseline but then again the baseline is for year 2000.

Thus, given that the goal is to conduct the study over a long time scale, 1950-2006 for climate oscillation impact, one of the critical pieces – crop production is one among many possible model simulation possibilities. I cannot be sure how close to reality the results really are.

I also do not understand what is being meant by food production. In the United States, Europe, China, Australia, etc. food production should include animal production. This has certainly not been done.

There is still a great deal of useful information in the modeling space and the methods used can trigger research in which observed crop production information is directly used. I think this result will find a much better avenue in a more specialist journal where crop-modeling results are presented.

Reviewer #2 (Remarks to the Author):

The manuscript addresses an interesting topic, which also represents an important gap in knowledge. In its current form, the methods and evidence presented do not match the major claims of the paper. I would only recommend publishing if: (a) claims are softened to be consistent with the rather weak proxy for food production used in the study, (b) evidence is presented about the goodness-of-fit between modeled production time series and production statistics in countries known to use good methodology, and (c) interpretation of results is better grounded in the literature on the teleconnections associated with each of the three climatic drivers.

- What are the major claims of the paper?

1) >2/3 of the world's major crop types are grown in areas where at least one of 3 climate oscillations (ENSO, IOD, NAO) is linked to significant anomalies in food production.

2) Identified possible new links between these oscillations and food production – especially for NAO – in Africa, Middle East and Southeast Asia.

- Are they novel and will they be of interest to others in the community and the wider field? If the conclusions are not original, it would be helpful if you could provide relevant references.

Good quality global studies of statistical association between agricultural production and climatic drivers are lacking in the literature, and are a welcome addition that would be of interest to the diverse community working at the interface between climate and agriculture.

- Is the work convincing, and if not, what further evidence would be required to strengthen the conclusions?

Unfortunately, the evidence is not adequate to support the main conclusions. This is because this study, like a few other recent global studies of drivers of agricultural production, used a proxy for crop production to make up for problems with national and sub-national agricultural production time-series statistics.

While this doesn't negate the value of the study, I see three major gaps between the conclusions

and the evidence. First, the manuscript does not provide evidence of the ability of the model to capture the year-to-year pattern of yields and production (see below). Therefore, the reader has no basis for relating modeled production to real production or assessing the major claims. Second, the manuscript seems to suggest a study of the relationship between three well-known climatic drivers and food production – which would be a welcome complement to analyses based on climate data. But in reality it seems to be an analysis of their relationship to monthly climate data aggregated in a new way (i.e., through a simulation model), and not analyses of influence of the three climatic drivers on an independent type of data. Third, the study intentionally filters out non-climatic drivers of variability of agricultural production, forcing 100% of the modeled production variability to be due to climate. Recent studies (e.g., Ray et al. (2015). *Nature Communications* 6) suggest that only about 1/3 of global crop yield variability is due to monthly climate variations. If the results over-state the role of climate in crop yield variability, it seems likely that they also overstate the variability due to the 3 climatic oscillations, and hence the areas where they have a significant influence on production.

In this case, simulation modeling was used to generate time series of yields for a set of “crop types” on a 0.5° grid, which were then aggregated to the Food Production Units reported in the manuscript. As I understand it, the only source of interannual variability of simulated yields is monthly CRU gridded data – which I believe is interpolated from whatever station observations were available – disaggregated to synthetic daily time series used to drive the simulation. Since it starts with monthly data, the methodology would not seem to capture any of the influence of, e.g., the timing and duration of dry spells, on production.

While the manuscript asserts that simulations match national production statistics available through FAO, I didn't see any goodness-of-fit statistics or graphical evidence of the degree to which year-to-year variations of simulated yields match available production statistics. The manuscript refers the reader to Kummu et al. (2014) *Hydrol. Earth Syst. Sci.* 18:447-461 to understand how the model was calibrated to match national production statistics. I also didn't see any evidence of goodness-of-fit in this paper, but a reference to another paper (Fader et al. (2010) *J. Hydrol.* 384:218-231) for details of how the model was made to fit agricultural production statistics. The Fader 2010 paper offers country-by-country, but not year-by-year, analysis of fit between modeled and reported production.

This is, in reality, an analysis of climatic teleconnections, between subnational climate variability and three widely recognized ocean temperature oscillations. It would be more convincing if it was more grounded in the available literature on where and how these teleconnections operate. This grounding in climate science is important, given the claims of new relationships that other studies haven't recognized, the claims of major shifts in the strengths and even directions of those relationships over time, and sometimes very strange spatial patterns of those relationships (e.g., Fig. 6a).

- On a more subjective note, do you feel that the paper will influence thinking in the field?

If published, it probably would influence thinking in the field, particularly among people coming at it from the agricultural sciences. If supported adequately and interpreted appropriately, this type of study would be quite welcome. However, given the limitations of the proxy used, the resulting likelihood of spurious relationships, and absence of discussion of physical mechanisms for the reported relationships, I'm concerned that it might have more influence among agricultural scientists than the evidence warrants.

- We would also be grateful if you could comment on the appropriateness and validity of any statistical analysis, as well the ability of a researcher to reproduce the work, given the level of detail provided.

There is not a lot of information about the bootstrap method used to test significance of

correlations and regressions. I assume it is valid for point-wise comparisons. I already noted the lack of statistical evidence relating simulation results to actual production.

Two other minor concerns:

First, using correlations based on a 21-year moving window as a basis for claiming major shifts in teleconnections seems rather weak. Since Pearson's correlation is heavily influenced by points near the tails of a distribution, I would expect that this method would show fluctuations in correlations in a long pair of time series that have a stationary correlation, as extreme values move into and out of the moving window. This aspect of the study seems to warrant more scrutiny, and explanations grounded in the theory behind the 3 oscillations and their teleconnections.

Second, since the analysis seems (as far as I can tell) to do point-wise tests of significance of correlations or regressions, and doesn't control for the multiplicity problem, it is likely that at least a few of the results are spurious. (I mentioned earlier about the more serious problem of making inferences about relationships between production and climatic drivers, based on simulation models that inflate the climatic component of production variability.)

Reviewer #3 (Remarks to the Author):

Three climate oscillation indices (ENSO, IOD and NAO) were linked with spatially-explicit global crop production during 1950–2006. Crop production data for maize, wheat rice and soybean were simulated with a global gridded crop model calibrated to reported trend-corrected FAO statistics. It was found that crop production in large parts of agriculture and global populations were affected by at least one index or in some cases a combination of indices.

This is an interesting study. While relationships of individual or combinations of indices with crop production have been carried before for specific regions and crops, this appears to be the first time where such a study has been conducted at a global scale for the main four global crops. The paper could be considered for publication after addressing the following points.

1. The quality of the simulated crops across the world should be shown in the Supplementary to allow the reader to judge how well the used model simulated yields.
2. I disagree. You can only 'predict' if an index is correlated to events into the future (e.g. next few months). You have shown the correlation of one or several indices to yield within a year, i.e. by the time you know the index you also know your yield. There is literature on using these climate indices for predicting yields well before the yield is known (shown via hindcasts). If you want to discuss predictions, the literature on forecasting using climate indices need to be brought in here (currently missing).
3. Some explanations on how these indices relate to climate drivers and therefore to growth and yield is needed. If, e.g. all indices relate to rainfall, then regions with a high proportion of irrigation (e.g. northern India) would be expected to be less related to such an index.
4. As crops are grown during parts of the year, it is not clear if an index was the average of a year or only for a specific period. This needs to be better explained and discussed.
5. It would be good to compare other regional studies to the results from the global study for these regions where similar indices were used. If relationships are similar would give confidence in the approach. The paper by Tian et al. 2015 in Agricultural and Forest Meteorology analyzed the relationship of three indices and simulated summer and winter crop yields in parts of a US. These relationships could be compared and discussed for the NAO index (common in both studies) for the common region in both studies and the two common crops.
6. It is not clear how a relationship of yield with an index can change from positive to negative over time. How can that be explained? Assuming an index is related to total rainfall amounts over

a period affecting yields, if sometimes the negative index gives the higher yields and sometimes the positive index, suggests that there is no relationship.

L233 change one three to one of three

L247 40 Billion seems not correct

L261 add crop before management

L284 spell out aren't

289-90 not clear why the same phase relates to increase and decrease

L316 why? Higher resolution will require more detailed inputs data which could also increase the error of the simulations

L334 what did they do for 'preparedness' which saved so much money? Be more specific.

Authors' responses on reviewers' comments

NCOMMS-16-22764: Two-thirds of global cropland area impacted by climate oscillations

We thank the editor and the reviewers for their careful evaluation of the manuscript and their constructive comments that helped us improve the manuscript considerably. We have taken all the comments carefully into consideration when revising the paper. The major revisions include: i) presenting the calibration results of the simulated yields and validation results against reported productivity and production, ii) assessing crop productivity ($\text{kcal ha}^{-1} \text{yr}^{-1}$) rather than crop production, and iii) adding detailed descriptions of the teleconnections, and how the current understanding is reflected in our results. Below are our detailed responses to the comments.

Reviewer #1 (Remarks to the Author):	
C1.1: This is basically a modeling study in which the authors related modeled crop production to observed climate oscillation indices. Ideally such studies should be conducted using the observed / reported crop production information. Irrespective of how sophisticated the methodology is, the fact remains that the result hinges on unverifiable crop production information.	R1.1: While we agree with the reviewer in the value of conducting this kind of studies using reported crop production information, we argue that using simulated crop yields brings additional information to these conventionally conducted studies. The main advantages of our simulation approach are as follows: - It is possible to assess the sole climate impact on crop yields. Reported crop yields include the impact of many other factors, making it difficult to differentiate the role of climate on yield variability- Time series of reported crop yields are usually available at country scale only and thus, by using simulated values, we can assess the impacts of total crop productivity at sub-national level. This is particularly important in large countries in which impacts of the climate oscillations may vary between different regions.- Reporting of yields and production are neither globally unified nor accurate for all countries, and by using simulations we can obtain comparable results across the globe. These points are now communicated in more detail in the Introduction (page 1, lines

	70-78). However, we acknowledge that in using modelled datasets it is important to first assess their ability to simulate the variables in question. Thus, we have now added a part dedicated to the calibration and validation of the model to the Methods section.
C1.2: How do the authors know that the simulated food production using LPJmL model is accurate? They have not been used to compare against historical crop yields, and the significantly more tricky production information (because to get the production correct one needs to accurately know the harvested area information as well). No amount of modeling can ever get the harvested extent correct, as that is discretion of farmers and thus only reported information can give an accurate picture. So, I am having difficulty in accepting the results other than that this is a modeling study so may or may not be accurate in each of the food production units.	R1.2: We regret that we did not communicate well the calibration and validation of the used model in the original manuscript in more details. Indeed, the model is calibrated against reported yields by FAO ¹ at country scale (see revised Methods, and new Supplementary Fig. 1), and subsequently validated against sub-national reported crop productivity of maize, rice, soybean and wheat from Ray et al ² as well as national scale reported productivity by FAO ¹ of all crop types included in the LPJmL model (see revised Methods, and new Supplementary Figs 2-7). In the revised version, we now present the calibration and validation results in detail. We have now:  - Switched our analysis time span to 1961-2010. This enables us to calibrate and validate the model with higher accuracy for the whole time frame of our analysis, as for example FAOSTAT data is available only since 1961. - We added a short section on the calibration and validation results (Section 2.1) and detail methods are described under the Methods section (Section 4.2). The detailed calibration and validation results of the model are shown in Supplementary Figs 2-7. - Calibration of the model (in terms of crop management processes) was conducted so that the modelled yields at country level match those of FAOSTAT as closely as possible. - In order to create a simulated productivity data set that is comparable to reported

	crop statistics for validation purposes, we did another simulation in which yields are decennially calibrated, similar to a part of a recent publication by Porkka et al.³.  - Validation of the model is conducted in four steps using two data sets of reported crop statistics: i) reported productivity of maize, rice, soybean and wheat at 0.5° resolution and ii) reported productivity of all crop types included in the LPJmL model. First, Spearman's correlation was calculated between simulated (decennially calibrated) and reported crop productivity. Spearman's correlation was chosen because of the very large variation among values. Second, Spearman's correlation was calculated between simulated (calibrated for year 2000) and reported (de-trended) crop productivity. Third, Spearman's correlation was calculated for the standard deviation and coefficient of variation (CV) between simulated (calibrated for year 2000) and reported (de-trended) crop productivity. See Supplementary Figs 2–7 for the results of the validation. Further, we conducted the sensitivity analysis (see new Methods) for the de-trended decennially calibrated data (Supplementary Fig. 15), and the results remain very similar compared to the original sensitivity analysis (Fig. 4). Indeed, the modelled production does not take into account actual harvested area (for reasons mentioned by the reviewer), but physical cropping areas when calculating the total production. Thus, we changed the focus of our analysis from crop production to crop productivity ($\text{kcal ha}^{-1} \text{yr}^{-1}$). We have added a description of this new procedure to the Methods section.
C1.3: Now the authors do indeed try to make the argument that the correct approach is indeed to use modeled crop yields (not production really because the authors can never simulate the harvested areas)	R1.3: We apologise for our poor expression. Our intention was to argue that in addition to climatological signals, observed yield variations also contain other signals, which

but land up contradicting themselves. If I understood the argument correctly, climate oscillations impact food production; but if observed data were used then the signal from climate oscillations would not be captured. And therein lies the contradiction - if the signal of climate oscillations is not present in the observed crop data that basically means that there is no impact – there is either really no impact or there is a coping mechanism.	may hide the signals related to climate oscillations. For example, Ray et al.² showed that 1/3 of global crop yield variability of wheat, maize, rice and soybean can be explained with climatological variations. Therefore, to obtain information about the potential impacts that climate oscillations have on crop productivity, isolating the climatological signal in crop productivity is required, especially if the goal is to quantify their effects. Furthermore, while we want to understand the spatially detailed impact of climate oscillations on crop productivity, reported information for all major crops is available only at the country level. But if country level yield data were used, oscillation signals could be dampened due to varying impacts within the country territories – particularly in case of large countries such as the USA, China or India.
C1.4: As I have noted above – production information requires two independent pieces of information – crop yields and crop harvested areas. The later is quite impossible to simulate. Even the former – crop yields from the LPJmL model has not really been validated to such a level of accuracy as to allow its use in isolating such a nuanced signal of climate oscillation on crop yields.	R1.4: We show now the calibration results of the model for crop yields (Supplementary Fig. 1) as well as validation results for crop productivity (Supplementary Figs 2-7), and overall the model performs very well. Therefore, although we agree that the model is a simplification of reality, we argue that its results bring additional, valuable insights on the impact of climate oscillations on crop productivity that would not be possible otherwise using only currently available reported information.
C1.5: There are several publications that was referred to show the modeled yields. I went through each of them to understand how well the modeled yields compared with reported yields as simulated by the author’s model of choice LPJmL. Fader et al., 2010 compared LPJmL simulated yields to FAO yields for year 1999 to 2003 with substantial	R1.5: Again, we apologise that the reviewer needed to search for model calibration and validation results from elsewhere. As stated above (R1.2), we paid careful attention to report and communicate these procedures (Section 4.2) and results (Supplementary Figs 1-7) in the revised manuscript and supplement.

errors in major crop countries even when simulating maize globally, and barley, rye, wheat, crops but only in temperate regions. So, I conclude that this paper's comparison / validation does not give me confidence to use for the period 1950-2006. Bondeau et al., 2007 simulated temperate cereals and maize and determined average yields over 1991 to 2000 and compared the simulated yields with FAO reports. So this report also cannot be used for conducting a study of food production from 1950 to 2006. Kummu et al., 2014 kept the food part of their study fixed at year 2000. Schaphoff et al., 2013 does not concern with crop yield / production but rather carbon using the LPJmL model. Gerten et al., 2011 held their crop production information fixed at year 2000. Rockström et al., 2009 was not concerned with simulating crop yields / production but on water availability. Rosenzweig et al., 2014 presented the results of intercomparing global crop model simulations out of which LPJmL is one model and all the models had disagreements in simulated crop yields to various amounts. Fader et al., 2013 is a study of future scenarios but does have a baseline but then again the baseline is for year 2000. Thus, given that the goal is to conduct the study over a long time scale, 1950-2006 for climate oscillation impact, one of the critical pieces – crop production is one among many possible model simulation possibilities. I cannot be sure how close to reality the results really are.	
C1.6: I also do not understand what is being meant by food production. In the United States, Europe, China, Australia, etc. food production should include animal production. This has certainly not been done.	R1.6: We have now changed our phrasing from food production to crop productivity (kcal ha⁻¹ yr⁻¹). The previous analyses included a simple component also for animal production, but this was omitted from the updated analyses. However, the crop productivity used here still includes animal feed.
C1.7: There is still a great deal of useful information in the modeling space and the methods used can trigger research in which observed crop production	R1.7: Thank you for the comment. Indeed, the main aim of this research is to identify hotspots where these oscillations have an

information is directly used. I think this result will find a much better avenue in a more specialist journal where crop-modeling results are presented.

effect on crop productivity, which could then trigger more specific studies. Given the global coverage with unprecedented spatial and temporal detail, providing significant new results, we are still confident that the study will be of interest to the broad readership of this journal.

Reviewer #2 (Remarks to the Author):	
C2.1: The manuscript addresses an interesting topic, which also represents an important gap in knowledge. In its current form, the methods and evidence presented do not match the major claims of the paper. I would only recommend publishing if: (a) claims are softened to be consistent with the rather weak proxy for food production used in the study, (b) evidence is presented about the goodness-of-fit between modeled production time series and production statistics in countries known to use good methodology, and (c) interpretation of results is better grounded in the literature on the teleconnections associated with each of the three climatic drivers.	R2.1: Thank you for the encouraging words and helpful comments. We briefly reply to your main points here, while also replying to each of your comment in detail below. a) In the revised manuscript, we: i) clarify what part of food production we refer to in the Methods section and altered the focus of our analysis to crop productivity ($\text{kCal ha}^{-1} \text{yr}^{-1}$); and ii) changed the analysis so that it only focuses on crop productivity, i.e. animal production is left out (although, animal feed is still included). b) We now present in detail the calibration and validation procedures of the model in Section 4.2 and results briefly under Section 2.1 and in details under Supplementary Figs 1-7; see response below. c) We added a more detailed discussion on the interpretation of the results in the main text, and a climatological comparison of our results with other studies in Supplementary Discussion 1.
C2.2: - What are the major claims of the paper? 1) >2/3 of the world's major crop types are grown in areas where at least one of 3 climate oscillations (ENSO, IOD, NAO) is linked to significant anomalies in food production. 2) Identified possible new links between these oscillations and food production – especially for NAO – in Africa, Middle East and Southeast Asia.	R2.2: These are still valid also after revisions.
C2.3: - Are they novel and will they be of interest to others in the community and the wider field? If the conclusions are not original, it would be helpful if you could provide relevant references. Good quality global studies of statistical association between agricultural production and climatic drivers are lacking in the literature, and are a welcome addition that would be of interest to the diverse community working at the	R2.3: Thank you for the comment. We agree and this gap initially triggered conducting this study.

interface between climate and agriculture.	
C2.4: Unfortunately, the evidence is not adequate to support the main conclusions. This is because this study, like a few other recent global studies of drivers of agricultural production, used a proxy for crop production to make up for problems with national and sub-national agricultural production time-series statistics. While this doesn't negate the value of the study, I see three major gaps between the conclusions and the evidence. First, the manuscript does not provide evidence of the ability of the model to capture the year-to-year pattern of yields and production (see below). Therefore, the reader has no basis for relating modeled production to real production or assessing the major claims. Second, the manuscript seems to suggest a study of the relationship between three well-known climatic drivers and food production – which would be a welcome complement to analyses based on climate data. But in reality it seems to be an analysis of their relationship to monthly climate data aggregated in a new way (i.e., through a simulation model), and not analyses of influence of the three climatic drivers on an independent type of data. Third, the study intentionally filters out non-climatic drivers of variability of agricultural production, forcing 100% of the modeled production variability to be due to climate. Recent studies (e.g., Ray et al. (2015). Nature Communications 6) suggest that only about 1/3 of global crop yield variability is due to monthly climate variations. If the results over-state the role of climate in crop yield variability, it seems likely that they also overstate the variability due to the 3 climatic oscillations, and hence the areas where they have a significant influence on production.	R2.4: We understand the reviewer's concerns and we have aimed to tackle those better in the revised manuscript. Indeed, the model is calibrated against reported yields by FAO¹ at country scale (see revised Methods, and new Supplementary Fig. 1), and subsequently validated against sub-national reported crop productivity of maize, rice, soybean and wheat from Ray et al² as well as national scale reported productivity by FAO¹ of all crop types included in the LPJmL model (see revised Methods, and new Supplementary Figs 2-7). See response to Reviewer #1 (cf. R1.2) for more details on this aspect. 1. In order to create a simulated productivity data set that is comparable to reported crop statistics for validation purposes, we did another simulation in which yields are decennially calibrated, similar to a part of a recent publication by Porkka et al.³. Validation of the model is conducted in four steps using two data sets of reported crop statistics: i) reported productivity of maize, rice, soybean and wheat at 0.5° resolution and ii) reported productivity of all crop types included in the LPJmL model. First, Spearman's correlation was calculated between simulated (decennially calibrated) and reported crop productivity. Spearman's correlation was chosen because of the very large variation among values. Second, Spearman's correlation was calculated between simulated (calibrated for year 2000) and reported (de-trended) crop productivity. Third, Spearman's correlation was calculated for the standard deviation and coefficient of variation (CV) between simulated (calibrated for year 2000) and reported (de-trended) crop productivity. See Supplementary Figs 2–7 for the results of the validation. Further, to evaluate whether adding other than climatological signals to

the simulated data affect the results, we conducted the sensitivity analysis (see Methods) for the de-trended decennially calibrated data (Supplementary Fig. 15), and the results remain very similar compared to the original sensitivity analysis (Fig. 4).

2. We are not sure if we fully understood the comment about the climate data, but clearly the data we used are an independent, external input to the model. Based on these climate data (as well as other input data and numerous interacting process representations), crop productivity and its spatio-temporal dynamics is calculated. This allows for quantification of the (isolated) effect of climate oscillations on crop productivity. Further, we updated the forcing data to daily data for the model simulations. We use now Global Meteorological Forcing Dataset for land surface modelling provided by Princeton University spanning the period 1948-2010 (whereby only data from 1961 were used here)⁴

3. Indeed, in addition to climatological signals, observed yield variations also contain other signals, which may hide the signals related to climate oscillations. Therefore, to obtain information about the potential impacts that climate oscillations have on crop productivity, isolating the climatological signal in crop productivity is somehow required, especially if the goal is to quantify their effects. The main aim of this research is to identify hotspots where these oscillations have an effect on crop productivity, which could then trigger more specific local studies. Given the global coverage with unprecedented spatial and temporal detail, providing significant new results, we are still confident that the study will be of value to the community working at the interface between climate and agriculture

C2.5: In this case, simulation modeling was used to generate time series of yields for a set of “crop types” on a 0.5° grid, which were then aggregated to the Food Production Units reported in the manuscript. As I understand it, the only source of interannual variability of simulated yields is monthly CRU gridded data – which I believe is interpolated from whatever station observations were available – disaggregated to synthetic daily time series used to drive the simulation. Since it starts with monthly data, the methodology would not seem to capture any of the influence of., e.g., the timing and duration of dry spells, on production.	R2.5: We agree that CRU is only but one global climate dataset with a rather crude (i.e. monthly) resolution, and that these data were stochastically disaggregated to get quasi-daily (and somewhat arbitrary) values. Therefore, we now updated the model forcing to the Global Meteorological Forcing Dataset for land surface modelling provided by Princeton University spanning the period 1948-2010 (whereby only data from 1961 were used here)⁴. This dataset provides daily meteorological forcing and thus enables a more accurate representation of climate influences on crop productivity. The update to the daily forcing data resolved some previously found strange patterns that appeared to be spurious, i.e. implausible in a climatological sense (e.g. strong influence of NAO to Southeast Asian crop productivity, now obsolete). But key results and conclusions remained the same.
C2.6: While the manuscript asserts that simulations match national production statistics available through FAO, I didn’t see any goodness-of-fit statistics or graphical evidence of the degree to which year-to-year variations of simulated yields match available production statistics. The manuscript refers the reader to Kummu et al. (2014) Hydrol. Earth Syst. Sci. 18:447-461 to understand how the model was calibrated to match national production statistics. I also didn’t see any evidence of goodness-of-fit in this paper, but a reference to another paper (Fader et al. (2010) J. Hydrol. 384:218-231) for details of how the model was made to fit agricultural production statistics. The Fader 2010 paper offers country-by-country, but not year-by-year, analysis of fit between modeled and reported production.	R2.6: Again, we apologise that the reviewer needed to search for model calibration and validation results from elsewhere. As stated above (see R2.4; R1.2), we paid careful attention to report and communicate these procedures (Section 4.2) and results (Section 2.1 and Supplementary Figs 1-7).
C2.7: This is, in reality, an analysis of climatic teleconnections, between subnational climate variability and three widely recognized ocean temperature oscillations. It would be more convincing	R2.7: We fully agree, and therefore added a detailed description of the teleconnections and how they operate, based on available literature, to the Supplementary Discussion

if it was more grounded in the available literature on where and how these teleconnections operate. This grounding in climate science is important, given the claims of new relationships that other studies haven't recognized, the claims of major shifts in the strengths and even directions of those relationships over time, and sometimes very strange spatial patterns of those relationships (e.g., Fig. 6a).	1. This is also now better reflected in the revised Discussion, providing more details on whether the correlations found here are supported also from a climatological viewpoint. See also the answer to the comment below regarding the analysis of potential changes in the linkages these oscillations and crop productivity.
C2.8: If published, [the paper] probably would influence thinking in the field, particularly among people coming at it from the agricultural sciences. If supported adequately and interpreted appropriately, this type of study would be quite welcome. However, given the limitations of the proxy used, the resulting likelihood of spurious relationships, and absence of discussion of physical mechanisms for the reported relationships, I'm concerned that it might have more influence among agricultural scientists than the evidence warrants.	R2.8: We believe that our study reveals important insights on the climatic impacts on crop productivity, as it is a first attempt to understand the impact of climate oscillations on past and present global crop productivity in a consistent framework. After the extensive revisions, we are confident that our conclusions are adequately supported.
C2.9: - We would also be grateful if you could comment on the appropriateness and validity of any statistical analysis, as well the ability of a researcher to reproduce the work, given the level of detail provided. There is not a lot of information about the bootstrap method used to test significance of correlations and regressions. I assume it is valid for point-wise comparisons. I already noted the lack of statistical evidence relating simulation results to actual production.	R2.9: We added more detailed explanations about the methodology of assessing the statistical significance of regression and correlation analyses to the Methods section, and we believe that it gives now better ground to reproduce the results.
C2.10: Two other minor concerns: First, using correlations based on a 21-year moving window as a basis for claiming major shifts in teleconnections seems rather weak. Since Pearson's correlation is heavily influenced by points near the tails of a distribution, I would expect that this method would show fluctuations in correlations in a long pair of time series that have a stationary correlation, as extreme values move into and out of the moving window. This aspect of the study seems to warrant more scrutiny,	R2.10: We decided to examine potential changes in the linkages between teleconnections and crop productivity, because it is well documented in the literature that the strength of, for example, ENSO has changed over time on timescales from millennia to decades⁵⁻⁸. Moreover, many studies have shown that its teleconnected influences on climates in distance regions have changed⁹⁻¹¹. For

and explanations grounded in the theory behind the 3 oscillations and their teleconnections.	example Ward et al.¹² revealed that there has also been a change in the correlation between ENSO and flood peak discharges over the last half century. Furthermore, a previous study indicates significant correlations between ENSO and water scarcity¹³. Therefore, we decided to investigate whether such an influence, and its change over time, exists with crop productivity as well. Furthermore, we agree with the reviewer that since Pearson's correlation requires calculating the mean of the variable in question, it is very sensitive anomalously high or low values. Thus, we changed the method of this analysis to Spearman's rank correlation, which linearizes the variable by calculating the Pearson's correlation of its ranks instead of the actual values. This transformation makes it less sensitive to non-linearity and extreme values. However, we decided to move this section to the Supplement to streamline the paper, as the results of this analysis are not within the main focus of the paper and additional analyses regarding the oscillations' climatological influences would've been required to make the analysis stronger.
C2.11: Second, since the analysis seems (as far as I can tell) to do point-wise tests of significance of correlations or regressions, and doesn't control for the multiplicity problem, it is likely that at least a few of the results are spurious. (I mentioned earlier about the more serious problem of making inferences about relationships between production and climatic drivers, based on simulation models that inflate the climatic component of production variability.)	R2.11: This is correct. By definition some of the correlations are likely spurious, since by setting a significance level, we accept a certain error rate. However, implicitly we control for this by using several different methods for our analyses. Further, in the main text, we control this by not looking at results of individual FPU, but patterns of several FPU.

Reviewer #3 (Remarks to the Author):	
C3.1: Three climate oscillation indices (ENSO, IOD and NAO) were linked with spatially-explicit global crop production during 1950-2006. Crop production data for maize, wheat rice and soybean were simulated with a global gridded crop model calibrated to reported trend-corrected FAO statistics. It was found that crop production in large parts of agriculture and global populations were affected by at least one index or in some cases a combination of indices. This is an interesting study. While relationships of individual or combinations of indices with crop production have been carried before for specific regions and crops, this appears to be the first time where such a study has been conducted at a global scale for the main four global crops. The paper could be considered for publication after addressing the following points.	R3.1: Many thanks for your encouraging comments and for reviewing our manuscript. We have addressed all the comments below. Especially, we want to point out that we now analyse the oscillations' impacts on crop productivity ($\text{kcal ha}^{-1} \text{yr}^{-1}$) (instead of total production kcal yr^{-1}) for 12 crop types that are included in the LPJmL model.
C3.2: 1. The quality of the simulated crops across the world should be shown in the Supplementary to allow the reader to judge how well the used model simulated yields.	R3.2: We agree that this information is needed. Model calibration and validation procedures and results are now presented briefly in a new Section 2.1 and in more detail under revised Methods (Section 4.2), while results are shown in details in new Supplementary Figs 1-7. See response to Reviewer #1 for more detail on this aspect (cf. R1.2).
C3.3: 2. I336 I disagree. You can only 'predict' if an index is correlated to events into the future (e.g. next few months). You have shown the correlation of one or several indices to yield within a year, i.e. by the time you know the index you also know your yield. There is literature on using these climate indices for predicting yields well before the yield is known (shown via hindcasts). If you want to discuss predictions, the literature on forecasting using climate indices need to be brought in here (currently missing).	R3.3: We agree that this was not the correct phrasing for describing our results in this section and thus rewrote the sentence. However, we do argue that there is predicative value in knowing the impacts these climate oscillations have on crop productivity, as their occurrence can be forecasted with lead-times ranging from several months up to a year¹⁴⁻¹⁷. Hence, providing information of crop productivity

	variations as well.
C3.4: 3. Some explanations on how these indices relate to climate drivers and therefore to growth and yield is needed. If, e.g. all indices relate to rainfall, then regions with a high proportion of irrigation (e.g. northern India) would be expected to be less related to such an index.	R3.4: We fully agree with this point. Therefore, we've added a detailed description of the teleconnections and how they operate, based on available literature, to the Supplementary Discussion 1. This is now also better reflected in the revised Discussion, providing more details on whether the correlations found here are supported also from a climatological viewpoint. Furthermore, we conducted a simulation in which we altered the irrigation set-up in the model and the results remained very similar (page 14, lines 432-339) and added a short statement of how temperature and rainfall affect crop growth in the model (page 13, lines 359-362).
C3.5: 4. As crops are grown during parts of the year, it is not clear if an index was the average of a year or only for a specific period. This needs to be better explained and discussed.	R3.5: The indices were transformed to yearly values by calculating the index means for those months where the oscillations tend to have the strongest signal (i.e. NDJ for ENSO, NDJF for NAO & SON for IOD). This is now more clearly explained in the Methods section and Supplementary Table 4.
C3.6: 5. It would be good to compare other regional studies to the results from the global study for these regions where similar indices were used. If relationships are similar would give confidence in the approach. The paper by Tian et al. 2015 in Agricultural and Forest Meteorology analyzed the relationship of three indices and simulated summer and winter crop yields in parts of a US. These relationships could be compared and discussed for the NAO index (common in both studies) for the common region in both studies and the two common crops.	R3.6: Thank you for pointing this out. We have now included additional comparisons to regional studies to Discussion, including the paper you suggested (page 11, line 305), which altogether corroborates our findings.
C3.7: 6. It is not clear how a relationship of yield with	R3.7: We decided to examine potential

an index can change from positive to negative over time. How can that be explained? Assuming an index is related to total rainfall amounts over a period affecting yields, if sometimes the negative index gives the higher yields and sometimes the positive index, suggests that there is no relationship.	changes in the linkages between teleconnections and crop productivity, because it is well documented in the literature that the strength of, for example, ENSO has changed over time on timescales from millennia to decades⁵⁻⁸. Moreover, many studies have shown that its teleconnected influences on climates in distance regions have changed⁹⁻¹¹. For example Ward et al.¹² revealed that there has also been a change in the correlation between ENSO and flood peak discharges over the last half century. Furthermore, a previous study indicates significant correlations between ENSO and water scarcity¹³. Therefore, we decided to investigate whether such an influence, and its change over time, exists with crop productivity as well. However, we decided to move this section to the Supplement to streamline the paper, as the results of this analysis are not within the main focus of the paper and additional analyses regarding the oscillations' climatological influences would've been required to make the analysis stronger.
C3.8: L233 change one three to one of three	R3.8: Thank you for the correction. This is now changed.
C3.9: L247 40 Billion seems not correct	R3.9: This is corrected.
C3.10: L261 add crop before management	R3.10: Added.
C3.11: L284 spell out aren't	R3.11: This is corrected.
C3.12: 289-90 not clear why the same phase relates to increase and decrease	R3.12: We apologise that this was poorly communicated. We have taken this point into account in rewriting this paragraph (page 10, lines 293-295).

C3.13: L316 why? Higher resolution will require more detailed inputs data which could also increase the error of the simulations	R3.13: This is correct. Because of the confusion this might cause as well as the vast amount of alternative paths this analysis could've taken, we decided to exclude this part from the updated manuscript.
C3.14: L334 what did they do for 'preparedness' which saved so much money? Be more specific.	R3.14: They prepared for floods that were expected to occur as a result of El Niño. They conducted flood prevention measures, such as distributing polypropylene bags and tarpaulin sheets to protect seeds and grains as well as provided financial support for other actions (e.g. filling of sandbags) alleviating the damage to crops. A short mention about this has been added to the Discussion (page 12, lines 346-347).

References

1. FAO. FAO Statistical Database. (2013).
2. Ray, D. K., Gerber, J. S., MacDonald, G. K. & West, P. C. Climate variation explains a third of global crop yield variability. *Nature communications* **6** (2015).
3. Porkka, M., Gerten, D., Schaphoff, S., Siebert, S. & Kummu, M. Causes and trends of water scarcity in food production. *Environmental Research Letters* **11**, 015001 (2016).
4. Sheffield, J., Goteti, G. & Wood, E. F. Development of a 50-year high-resolution global dataset of meteorological forcings for land surface modeling. *J. Clim.* **19**, 3088-3111 (2006).
5. Li, J. *et al.* El Niño modulations over the past seven centuries. *Nature Climate Change* **3**, 822-826 (2013).
6. McPhaden, M. J., Zebiak, S. E. & Glantz, M. H. ENSO as an integrating concept in earth science. *Science* **314**, 1740-1745 (2006).
7. Cane, M. A. The evolution of El Niño, past and future. *Earth Planet. Sci. Lett.* **230**, 227-240 (2005).
8. Mann, M. E., Cane, M. A., Zebiak, S. E. & Clement, A. Volcanic and solar forcing of the tropical Pacific over the past 1000 years. *J. Clim.* **18**, 447-456 (2005).
9. Dettinger, M. D., Cayan, D. R., McCabe, G. J. & Marengo, J. A. in *Multiscale streamflow variability associated with El Niño/Southern oscillation* (Cambridge University Press, 2000).
10. McCABE, G. J. & Dettinger, M. D. Decadal variations in the strength of ENSO teleconnections with precipitation in the western United States. *Int. J. Climatol.* **19**, 1399-1410 (1999).
11. Gershunov, A. & Barnett, T. P. Interdecadal modulation of ENSO teleconnections. *Bull. Am. Meteorol. Soc.* **79**, 2715-2725 (1998).
12. Ward, P. J., Eisner, S., Flörke, M., Dettinger, M. D. & Kummu, M. Annual flood sensitivities to El Niño–Southern Oscillation at the global scale. *Hydrology and Earth System Sciences* **18**, 47-66 (2014).
13. Veldkamp, T. I., Eisner, S., Wada, Y., Aerts, J. C. & Ward, P. J. Sensitivity of water scarcity events to ENSO-driven climate variability at the global scale. (2015).
14. Ludescher, J. *et al.* Very early warning of next El Niño. *Proc. Natl. Acad. Sci. U. S. A.* **111**, 2064-2066 (2014).
15. Scaife, A. *et al.* Skillful long-range prediction of European and North American winters. *Geophys. Res. Lett.* **41**, 2514-2519 (2014).
16. Ludescher, J. *et al.* Improved El Niño forecasting by cooperativity detection. *Proc. Natl. Acad. Sci. U. S. A.* **110**, 11742-11745 (2013).

17. Luo, J., Behera, S., Masumoto, Y., Sakuma, H. & Yamagata, T. Successful prediction of the consecutive IOD in 2006 and 2007. *Geophys. Res. Lett.* **35** (2008).

Reviewers' comments:

Reviewer #1 (Remarks to the Author):

I would like to thank the authors for agreeing to make major changes based on my comments. The revised draft reflects the challenges that process-based crop modelers face in simulating crop yields correctly. Analyzing the simulated crop yields to reported yields was necessary because only after convincingly simulating the crop productivity we can move ahead and draw conclusions on the relationship between global scale climate oscillations and crop yields.

In the previous round of review I raised the issue that simulated crop yield may not reflect the reported yields (which may also not accurately reflect real yields everywhere due to data reporting errors). Including Supplementary Figures 2 and 3 starts to address this issue.

I started by looking at the comparison of the simulated yields with the FAO reported yields more closely. In the methods section the authors suggested that "correlations are very high", but when I looked at Supplementary Figure 3a (which had higher correlation numbers compared to other correlation maps i.e. Supplementary Figures 2 and 3b) many countries are actually in the 0.3 to 0.5 correlation range and one – Chad – is even negatively correlated. Moreover many countries such as DRC, Russian Federation etc. are missing in Supplementary Figure 3, but present in Supplementary Figure 2 and conclusions are drawn over all areas (Figures 4 & 5).

Even though simulated versus reported yields do not match well everywhere, such as negatively correlated regions, and regions with low correlations (below 0.5) the authors however used all the areas to determine the strength of climate oscillations on crop yields. The authors could give more thought to: if model simulated data does not match well with independent but reported numbers in certain regions / countries of the world (against which the model is calibrated) should they proceed to study the model reported number. The task is made more difficult because I had no way of figuring out whether it was cassava or sorghum yields not correlating well, or maize yields, for example in Africa. Also I would have expected uniformly high correlation / match of crop yields in all the countries of Western Europe and between European, USA, Brazil and Australian crop yields.

With correlations of modeled versus reported yields in the range of 0.5, and correlations between oscillations and yield in the range of 0.4, is it possible that the authors are getting the wrong signals.

I wanted to state that there are numerous global scale crop modelers & modeling groups – for example the AgMIP team – groups using the DSSAT and APSIM crop models etc. The report in the present form also needs to be reviewed by crop modeling experts to comment upon the LPJmL simulations. None of the 3 reviewers appeared to have commented on the LPJmL model and its suitability to simulate crop yields versus other crop models – in other words would this work be better performed using DSSAT for example.

Reviewer #2 (Remarks to the Author):

I don't have time to read in detail. From the authors' response to reviewers' comments, and scanning a few sections, I'm satisfied that they addressed the major issues.

Reviewer #3 (Remarks to the Author):

The authors have attended to most of the comments and improved the manuscript. There are some points still need to be addressed.

Main points:

L344 this is not correct. The paper did not show a forecast. The analysis correlated yields with climate indices. This is not forecasting! Please correct.

L388 be specific. What parameters did you adjust and why?

L396 not sure how you calibrated to land use and CO2. These things are given inputs and should not be changed (adjusted) to make your simulations fit better. That is not scientific!

Minor points

L118 change is to was

L119 delete very

L122 significant results of what? Or do you mean significant correlation? Please correct

L150 delete a

L255 replace influence with 'from other factors'

L262 add after 'compensated' 'through other not-affected regions and hence by...' [delete 'for example']

L263 replace 'deficiency shocks' with 'low productivity years'

L265 replace losses with production

L267 replace portion with part

L268 not clear. What are natural occurring droughts and how do they differ from the droughts related to the indices? Do you mean socio-economic factors here?

L278 replace in with at

L305 replace lags with lag time

L312 delete very

L347-348 supply some of the details here you gave in the response to the earlier comment.

L351 delete 'many of'

L400 what was the outcome? Give some details.

L414 delete very

L420 delete very

L425 give reasons why?

L429 replace the with a

L436 impact on what?

L440 replace remain with are

L440 delete very

L440 change similarfor to 'similar for'

Authors' responses on reviewers' comments

NCOMMS-16-22764: Two-thirds of global cropland area impacted by climate oscillations

We wish to thank the editor and the reviewers for their careful evaluation of the manuscript and their constructive comments that helped us improve the manuscript considerably. We have taken all the comments into consideration when revising the paper. The major revisions include: i) a literature review of the skill of LPJmL in simulating yields of varying crops, ii) supplementary figures showing correlations between simulated and reported productivity of the 12 crop types separately at country-level and iii) masking the results in areas where skill is low, specifically where correlations between simulated and reported productivity are insignificant or negative. These revisions strengthen the argument that the model is a suitable choice for this study (both on its own merits and compared to other models), and that our conclusions adequately reflect the uncertainty involved. The article provides a transparent representation of the skill of the model and clearly differentiates which outputs are likely to be most robust.

Reviewer #1 (Remarks to the Author):	
C1.1: I would like to thank the authors for agreeing to make major changes based on my comments. The revised draft reflects the challenges that process-based crop modeler face in simulating crop yields correctly. Analyzing the simulated crop yields to reported yields was necessary because only after convincingly simulating the crop productivity we can move ahead and draw conclusions on the relationship between global scale climate oscillations and crop yields. In the previous round of review I raised the issue that simulated crop yield may not reflect the reported yields (which may also not accurately reflect real yields everywhere due to data reporting errors). Including Supplementary Figures 2 and 3 starts to address this issue. I started by looking at the comparison of the simulated yields with the FAO reported yields more closely. In the methods section the authors suggested that “correlations are very high”, but when I looked at Supplementary Figure 3a (which had higher correlation numbers compared to other correlation maps i.e. Supplementary Figures 2 and 3b) many countries are actually in the 0.3 to 0.5 correlation range and one – Chad – is even negatively correlated.	R1.1: Thank you for the overall positive view on our revisions. We agree with the reviewer that it is critical that we carefully qualify our findings on the impacts of these climate oscillations in areas where we did not find significant positive correlation between reported and simulated crop productivity. Therefore, when revising the manuscript:  - We identified the areas where the correlation between decennially calibrated and reported productivity is insignificant or negative at country-scale, i.e. using FAOSTAT data. The correlations were calculated with maximum number of available years, while limiting the minimum number of available years to 15 (Supplementary Figs 6 and 7). This allowed us to conduct these comparisons also for e.g. countries of former Soviet Union. The 15 year cut off (actually, the remaining countries had either 11 years or 5 years of data available) was applied

Moreover many countries such as DRC, Russian Federation etc. are missing in Supplementary Figure 3, but present in Supplementary Figure 2 and conclusions are drawn over all areas (Figures 4 & 5).	to avoid significance test errors due to small sample size.  - In all map figures (Figs 1, 3-5, Supplementary Figs 22-24) we masked these areas with striped colouring while in the aggregated results we either provide results for the masked results only (e.g. Figure 2) or for both masked and non-masked results separately (e.g. Table 1 shows masked results while Supplementary Table 4 unmasked ones). - We agree that in Methods the correlation results were described too superficially. We have now described the correlations in a more detailed manner (Lines 440-445). - In Discussion we ensure that we draw conclusions only from those areas where simulated productivity is found to significantly correlate with reported data.
C1.2: Even though simulated versus reported yields do not match well everywhere, such as negatively correlated regions, and regions with low correlations (below 0.5) the authors however used all the areas to determine the strength of climate oscillations on crop yields. The authors could give more thought to: if model simulated data does not match well with independent but reported numbers in certain regions / countries of the world (against which the model is calibrated) should they proceed to study the model reported number. The task is made more difficult because I had no way of figuring out whether it was cassava or sorghum yields not correlating well, or maize yields, for example in Africa. Also I would have expected uniformly high correlation / match of crop yields in the all the countries of Western Europe and between European, USA, Brazil and Australian crop yields.	R1.2: As stated above (see R1.1), we have masked all of the results in areas where the correlation between decennially calibrated and reported productivity is either insignificant or negative or data is lacking. Instead of using a 0.5 correlation coefficient threshold as suggested by the reviewer, we used $p < 0.1$ to determine whether the correlation is significant. We believe this test is more appropriate as correlations can be i) low but significant due data errors, or ii) high but insignificant in short time series. To enable a reader to differentiate LPJmL's performance in simulating the productivity of each crop type, we have added country-level comparisons between simulated and reported productivity for each crop type separately (Supplementary Figs 10 and 11). Further, in order to demonstrate that the calibration of the yields is successful also for

	the other decades (for the decennially calibrated productivity), we have included the calibration results (scatter plots as well as corresponding maps) for each crop type for 1981–1990 (Supplementary Figs 2-5).
C1.3: With correlations of modeled versus reported yields in the range of 0.5, and correlations between oscillations and yield in the range of 0.4, is it possible that the authors are getting the wrong signals.	R1.3: We have now masked the potential areas where there is low model performance, thus informing the reader to pay caution to results in these areas. Please see also R1.1 and R1.2 for more details.
C1.4: I wanted to state that there are numerous global scale crop modelers & modeling groups – for example the AgMIP team – groups using the DSSAT and APSIM crop models etc. The report in the present form also needs to be reviewed by crop modeling experts to comment upon the LPJmL simulations. None of the 3 reviewers appeared to have commented on the LPJmL model and its suitability to simulate crop yields versus other crop models – in other words would this work be better performed using DSSAT for example.	R1.4: Similarly to pDSSAT and pAPSIM, LPJmL is also part of AgMIP, as well as ISIMIP (The Inter-Sectoral Impact Model Intercomparison Project). We have now included three paragraphs about the performance of LPJmL in simulating crop yields into Discussion (Lines 326–356), e.g. drawing on very recent (published 2017) articles by Müller et al.¹, Frieler et al.² and Schauberger et al.³. In general, LPJmL performs well in these studies (see e.g. Figs 1-4 in Frieler et al.² and Figs 1-4 in Müller et al.¹). Among the 14 models in AgMIP, LPJmL ranks first in simulating wheat yields when simulation results are compared to reported yields (Fig. 2 and S43 in Müller et al.¹). Compared to other gridded global crop models, it has also been noted to skilfully simulate potential future maize, wheat, soybean and rice yields⁴ as well as the responses to temperature and drought stress³. The two models suggested by the reviewer (pDSSAT and pAPSIM) are parameterized to simulate maize, soybean and wheat, while pDSSAT also simulates rice¹ and pAPSIM simulates millet and sorghum⁵. In contrast, our study considers 8/7 major crop types in addition to these 4/5 crops. Thus, reproducing the data used in this study with

pDSSAT or pAPSIM would not be possible.

In the intercomparison conducted by Müller et al.¹, LPJmL out-performs both of these models in wheat simulations and pDSSAT in simulating rice (Figs 3 & S44 in Müller et al.¹). In maize simulation skill, they come very close, LPJmL being second after pDSSAT (Figs 1 & 11 in Müller et al.¹). For soybean simulations, their rankings strongly depend on initial simulation setup (Fig. S45 in Müller et al.¹), while correlations remain significant for all models in a globally aggregated comparison setup (Fig. 4 Müller et al.¹). pDSSAT shows abnormally high (compared to other models and observations) fluctuations in yield variability for all the crops simulated (Fig 4 in Frieler et al.²). Finally, patterns of the maximum fraction of yield variability that can be attributed to weather shown in Frieler et al.² (Fig S16) strongly resemble the results of the evaluation conducted in this study (cf. Supplementary Fig. 11).

Further, we stress that the reported crop statistics against which the results from LPJmL and the other GGCMs are compared are highly uncertain in some regions. For example, Müller et al.¹ found that the two reference data sets of reported crop statistics they exploited did not correlate well everywhere. In addition, reported crop yields usually refer to the production on the area harvested rather than planted, ignoring effects of complete crop failure and potentially underestimating the variability in crop yields caused by climate extremes.

Given that a third of global crop yield variability can be attributed to climate variability⁶ and that temporal dynamics in crop productivity simulated with LPJmL are here driven by only climatological input data, it is not to be expected that it reproduces all observed crop yield dynamics.

--	--

Reviewer #2 (Remarks to the Author):	
C2.1: I don't have time to read in detail. From the authors' response to reviewers' comments, and scanning a few sections, I'm satisfied that they addressed the major issues.	R2.1: Thanks for this general positive view on our revision. We thank you for the comments and suggestions for improving this study in the previous review round.

Reviewer #3 (Remarks to the Author):	
C3.1: L344 this is not correct. The paper did not show a forecast. The analysis correlated yields with climate indices. This is not forecasting! Please correct.	R3.1: Indeed, we did not show a forecast ourselves. However, what we argue is that improved understanding of the effects of climate oscillations, as demonstrated in our study, will add value to short-term forecasts if considered in such approaches. We have now rephrased this sentence to communicate our general point better (Lines 363-365).
C3.2: L388 be specific. What parameters did you adjust and why?	R3.2: The calibration of agronomic practices was conducted by adjusting three parameters for each crop type and country: 1) the maximum leaf area index, with values between 1 and 7, 2) the harvest index, i.e. the maximal fraction of above-ground biomass allocated to storage organs at harvest in the absence of water stress, and 3) the radiation use efficiency, i.e. efficiency to convert intercepted photosynthetically active radiation into biomass). The calibration aimed for simulated yields to best match country-level crop yields reported for year 2000. Not all management interventions are currently being modelled (incl. fertilizer application), making these adjustments necessary in order to ensure adequate simulation of crop yields. These points have now been added to the text (Lines 406-413).
C3.3: L396 not sure how you calibrated to land use and CO₂. These things are given inputs and should not be changed (adjusted) to make your simulations fit better. That is not scientific!	R3.3: We did not calibrate to land use or CO₂ concentration. Instead, the calibration of yields in LPJmL is conducted by only adjusting three specific parameters as explained above (R3.2). Land use and CO₂ are handled purely as inputs. We have now revised the text to avoid misunderstandings (Lines 427-429).

C3.4: L118 change is to was	R3.4: Corrected as suggested.
C3.5: L119 delete very	R3.5: Corrected as suggested.
C3.6: L122 significant results of what? Or do you mean significant correlation? Please correct	R3.6: We changed the location of the whole paragraph and now explain the correlations in a more detailed manner (Lines 440-455).
C3.7: L150 delete a	R3.7: Corrected as suggested.
C3.8: L255 replace influence with 'from other factors'	R3.8: Corrected, but instead of 'factors', we used 'drivers' (Lines 254-256).
C3.9: L262 add after 'compensated' 'through other not-affected regions and hence by...' [delete 'for example']	R3.9: Changed partly as suggested: 'This suggests that regional crop production deficits due to these oscillations could be compensated by interregional trade from non-affected regions.'
C3.10: L263 replace 'deficiency shocks' with 'low productivity years'	R3.10: Corrected as suggested.
C3.11: L265 replace losses with production	R3.11: Corrected as suggested.
C3.12: L267 replace portion with part	R3.12: Corrected as suggested.
C3.13: L268 not clear. What are natural occurring droughts and how do they differ from the droughts related to the indices? Do you mean socio-economic factors here?	R3.13: Corrected and clarified (Lines 268-269). Changed 'naturally occurring droughts' to 'droughts occurring due to other factors'.

C3.14: L278 replace in with at	R3.14: Corrected as suggested.
C3.15: L305 replace lags with lag time	R3.15: Corrected as suggested.
C3.16: L312 delete very	R3.16: Corrected as suggested.
C3.17: L347-348 supply some of the details here you gave in the response to the earlier comment.	R3.17: Corrected as suggested (Lines 265-269). The sentence now states: “The FAO provided a global action plan to tackle agricultural vulnerability to the 2015–2016 El Niño event, and for example in Somalia, the preparedness towards the El Niño of 2016 prevented crop losses worth millions of dollars by actions (e.g. polypropylene bag and tarpaulin sheet distribution) allowing farmers to prepare for expected flooding.”
C3.18: L351 delete ‘many of’	R3.18: Corrected as suggested.
C3.19: L400 what was the outcome? Give some details.	R3.19: The outcomes are now explained in a following paragraph: Lines 440-455.
C3.20: L414 delete very	R3.20: Corrected as suggested.
C3.21: L420 delete very	R3.21: Corrected as suggested.
C3.22: L425 give reasons why?	R3.22: CV is larger for reported than simulated crop productivity in Southern Africa, potentially because many non-climatological factors influence crop productivity fluctuations in those areas.

C3.23: L429 replace the with a	R3.23: Corrected as suggested.
C3.24: L436 impact on what?	R3.24: To clarify the sentence, we changed 'impact' to 'effect' (Lines 463-466).
C3.25: L440 replace remain with are	R3.25: Corrected as suggested.
C3.26: L440 delete very	R3.26: Corrected as suggested.
C3.27: L440 change similarfor to 'similar for'	R3.27: Corrected as suggested.

References

1. Müller, C. *et al.* Global gridded crop model evaluation: benchmarking, skills, deficiencies and implications. *Geoscientific Model Development* **10**, 1403 (2017).
2. Frieler, K. *et al.* Understanding the weather signal in national crop-yield variability. *Earth's Future* (2017).
3. Schauburger, B. *et al.* Consistent negative response of US crops to high temperatures in observations and crop models. *Nature communications* **8**, 13931 (2017).
4. Rosenzweig, C. *et al.* Assessing agricultural risks of climate change in the 21st century in a global gridded crop model intercomparison. *Proc. Natl. Acad. Sci. U. S. A.* **111**, 3268-3273 (2014).
5. <https://www.isimip.org/impactmodels/>.
6. Ray, D. K., Gerber, J. S., MacDonald, G. K. & West, P. C. Climate variation explains a third of global crop yield variability. *Nature communications* **6** (2015).

REVIEWERS' COMMENTS:

Reviewer #1 (Remarks to the Author):

The authors have addressed my concerns. I hope that the authors will agree that this improved the work overall. I hope that this work will be useful for planning purposes and trigger further research.

Authors' responses on reviewers' comments

NCOMMS-16-22764: Two-thirds of global cropland area impacted by climate oscillations

We are delighted to learn that the reviewers were satisfied with our replies and revisions. We want to thank all the reviewers for their comments and suggestions for improving the quality of this study. Please find our responses to the reviewer's comments below.

Reviewer #1 (Remarks to the Author):	
The authors have addressed my concerns. I hope that the authors will agree that this improved the work overall. I hope that this work will be useful for planning purposes and trigger further research.	We want to thank Reviewer #1 for a critical evaluation of this study. Indeed, the comments and suggestions provided by Reviewer #1 improved the study considerably.